# Experimental and mathematical insights on the interactions between poliovirus and a defective interfering genome

Yuta Shirogane[1,2☯], Elsa Rousseau[3,4☯], Jakub Voznica[1,5☯], Yinghong Xiao[1], Weiheng Su[1,6], Adam Catching[1], Zachary J. Whitfield[1], Igor M. Rouzine[1,7¤], Simone Bianco[3,4]*, Raul Andino[1]*

**1** Department of Microbiology and Immunology, University of California, San Francisco, California, United States of America, **2** Department of Virology, Faculty of Medicine, Kyushu University, Fukuoka, Japan, **3** Department of Industrial and Applied Genomics, AI and Cognitive Software Division, IBM Almaden Research Center, San Jose, California, United States of America, **4** NSF Center for Cellular Construction, University of California, San Francisco, California, United States of America, **5** ENS Cachan, Université Paris-Saclay, Cachan, France, **6** National Engineering Laboratory for AIDS Vaccine, School of Life Sciences, Jilin University, Changchun, China, **7** Laboratoire de Biologie Computationnelle et Quantitative, Sorbonne Universite, Institut de Biologie Paris-Seine, Paris, France

☯ These authors contributed equally to this work.
¤ Current address: Immunogenetics, Sechenov Institute of Evolutionary Physiology and Biochemistry RAS, Saint-Petersburg, Russia
* sbianco@us.ibm.com (SB); raul.andino@ucsf.edu (RA)

**Data Availability Statement:** All relevant data are within the manuscript and its Supporting information files.

## Abstract

During replication, RNA viruses accumulate genome alterations, such as mutations and deletions. The interactions between individual variants can determine the fitness of the virus population and, thus, the outcome of infection. To investigate the effects of defective interfering genomes (DI) on wild-type (WT) poliovirus replication, we developed an ordinary differential equation model, which enables exploring the parameter space of the WT and DI competition. We also experimentally examined virus and DI replication kinetics during co-infection, and used these data to infer model parameters. Our model identifies, and our experimental measurements confirm, that the efficiencies of DI genome replication and encapsidation are two most critical parameters determining the outcome of WT replication. However, an equilibrium can be established which enables WT to replicate, albeit to reduced levels.

## Author summary

We used a combination of mathematical modeling and experimental measurements to better understand the interaction between poliovirus (WT) and defective viral genomes (DI) co-infecting a single cell. We developed an ordinary differential equations (ODEs) model that capture the dynamics of the interaction, and provides a quantitative evaluation of the parameters, and their interactions, affecting virus replication. We observed that WT and DI compete directly for capsid proteins during encapsidation and for shared resources necessary for replication, a phenomenon known as exploitation competition.

**Funding:** The Defense Advanced Research Projects Agency founded the project (HR0011-17-2-0027 to RA and SB). IMR is funded by Agence National de Recherche, France, grant J16R389. The funders had no role in study design, data collection and analysis, decision to publish, or preparation of the manuscript.

**Competing interests:** The authors have declared that no competing interests exist.

Hence, our study identifies two different types of competition occurring during the co-infection of a cell by WT and DI.

## Introduction

Co-infections, the simultaneous infection of a host by multiple pathogen species, are frequently observed [1, 2]. The interactions between these microrganisms can determine the trajectories and outcomes of infection. Indeed, competition between pathogen species or strains is a major force driving the composition, dynamics and evolution of such populations [1, 3]. Three types of competition among free-living organisms have been defined from an ecological point of view: exploitation, apparent, and interference competition [1–3]. Exploitation competition is a passive process in which pathogens compete for access to host resources. Apparent competition occurs when the a population increases in the number of individuals. This, in turn, results in the increase in number of predators. In this scenario, other group of individuals can be indirectly affected by the the increase in the number of predators hunting both groups in the area. Thus, competition that is not due to using shared resources, but to having a predator in common [4]. In the context of the ecology of infection, apparent competition is associated with the stimulation of host immune responses, which acts as predator [3]. Interference competition represents a direct attack inhibiting the growth, reproduction or transmission of competitors, either chemically or mechanically [5].

In the current study we aim to understand better the interference competition process between two RNA virus genomes co-infecting a single cell. Thus, we examine the competition between full length wild-type (WT) poliovirus type 1 (PV1) and a replication competent replicon RNA derived from poliovirus type 1. This replicon was engineered by deleting the entire region encoding for capsid proteins. Poliovirus, the causative agent of poliomyelitis, is a positive-sense single-stranded RNA enterovirus belonging to the family *Picornaviridae* [6]. Upon cell infection, WT poliovirus initiates a series of processes that leads to the production of structural and nonstructural viral proteins and genome amplification. Structural proteins encapsidate the viral RNA, which leads to infectious viral particle production and cell-to-cell spread. Indeed, only encapsidated poliovirus genomes can survive outside the cells and can bind to new cells to initiate infections.

As an RNA virus, poliovirus is characterized by a high level of genome plasticity and evolution capacity, due to both high replication rate and error-prone nature of viral RNA polymerase [7, 8], which generates a large proportion of mutants in the viral population, called viral quasispecies [9–11]. In addition, defective genomes, carrying genomic RNA deletions, are thought to be produced also by RNA polymerase errors during RNA replication [12–14]. Natural poliovirus DI particles have been observed carrying in-frame deletions within the P1 region of the genome encoding structural capsid proteins [15, 16]. The DI genome used in this study features similar deletion of the P1 region, while expression of nonstructural proteins is not affected. When co-infecting cells with the WT, it can exploit capsid proteins produced by the WT to form DI particles, and in this way spread to new cells. However, it is not able to reproduce progeny in the absence WT helper virus [16].

From an ecological perspective, the WT can be viewed as a cooperator, producing capsid proteins as public goods. The DI particles are non-producing cheaters, bearing no production cost while exploiting the capsid protein products from the WT [1, 3]. Hence, co-infection should enable DI particles to replicate and spread, while hindering WT growth and propagation by interference competition.

DI particles have been identified in a number of viral species, such as vesicular stomatitis virus [17], poliovirus [18], ebola virus [19], dengue virus [20] or influenza virus [21, 22]. Given that DI particles compete and reduce WT production, they can modulate the outcome of infection [18, 22]. A recent study showed that defective viral genomes can contribute to attenuation in influenza virus infected patients [23], and they can protect from experimental challenges with a number of pathogenic respiratory viruses [24, 25].

Several studies have examined the interactions between defective particles and their parent viruses using a mathematical formalism. The majority of the prior work has focuses on the intercellular competition process in various settings (bioreactor production systems, passaging experiments, etc.) [26–28]. Much less attention has been devoted to the investigation of the intracellular dynamic of the WT-DI system. Stauffer Thompson and colleagues [29] investigate the impact of DIP dosing on the production of new DIP during a WT-DIP co-infection in Vesicular Stomatitis Virus. Differently from the model we present, their model does not explicitly take into account the competition for cellular resources. Laske and colleagues introduce a very detailed kinetic model of the dynamic of influenza with its DIs. [30]. While similar intracellular kinetic models exist for poliovirus [31], their application for the biological scope of this paper brings an unnecessary burden from the point of view of computation and parameter estimation.

In this study, we examined the competition between poliovirus WT and DI genomes within cells during one infection cycle, using a coarse grained dynamic model. DI particles are spontaneously generated during a significant number of virus infection, but the factors affecting DIs generation and propagation are not well understood. We then developed an ordinary differential equations (ODEs) mathematical model that captures the dynamics of DI and WT replication and encapsidation. We aimed to understand the processes within a single replication cycle that determine interference of the defective genome with WT within a cell. Our study indicates that DI and WT genomes compete for limiting cellular resources required for their genome amplification and for capsid proteins required for particle morphogenesis. The model was further used to evaluate the potential outcome of the interaction between DIs and WT viruses over a large range of parameter values and initial conditions (multiplicities of infection, temporal spacing and order of infection). Using DI variants carrying mutations that change their kinetic of replication, we further examine the predictive value of our model and its value to improving DI designs. Thus, the mathematical model described facilitate the understanding and interpretation of the biological impact of DI particles in the context of virus infection.

## Results

### Interference of WT poliovirus production by DI genomes

Initially, we evaluated whether DI genomes, carrying a deletion of the entire region encoding for capsid proteins, could affect WT virus replication (Fig 1A). The DI genome used in this study does not produce capsid proteins and is thus unable to encapsidate its genome and spread to other cells. However, it retains full capacity to produce nonstructural viral proteins and replicate its genomic RNA. WT poliovirus and DI genomic RNAs were transfected by electroporation to HeLaS3 cells and infectious titers of WT virus were determined over time by plaque assay (see Material and methods). As a control we used a replication-incompetent defective RNA lacking the capsid-encoding region, a part of 3D-polymerase encoding region, and the entire 3' nontranslated region (NTR). HeLa S3 cells transfected by only WT genomes produced nearly $1 \times 10^7$ PFU/ml WT virus 9 hours after transfection, while co-transfection of WT genomes together with DI RNAs at a ratio WT:DI = 1:4 resulted in 100-fold decrease of WT titers (Fig 1B). The non-replicating defective RNA did not affect WT virus production,

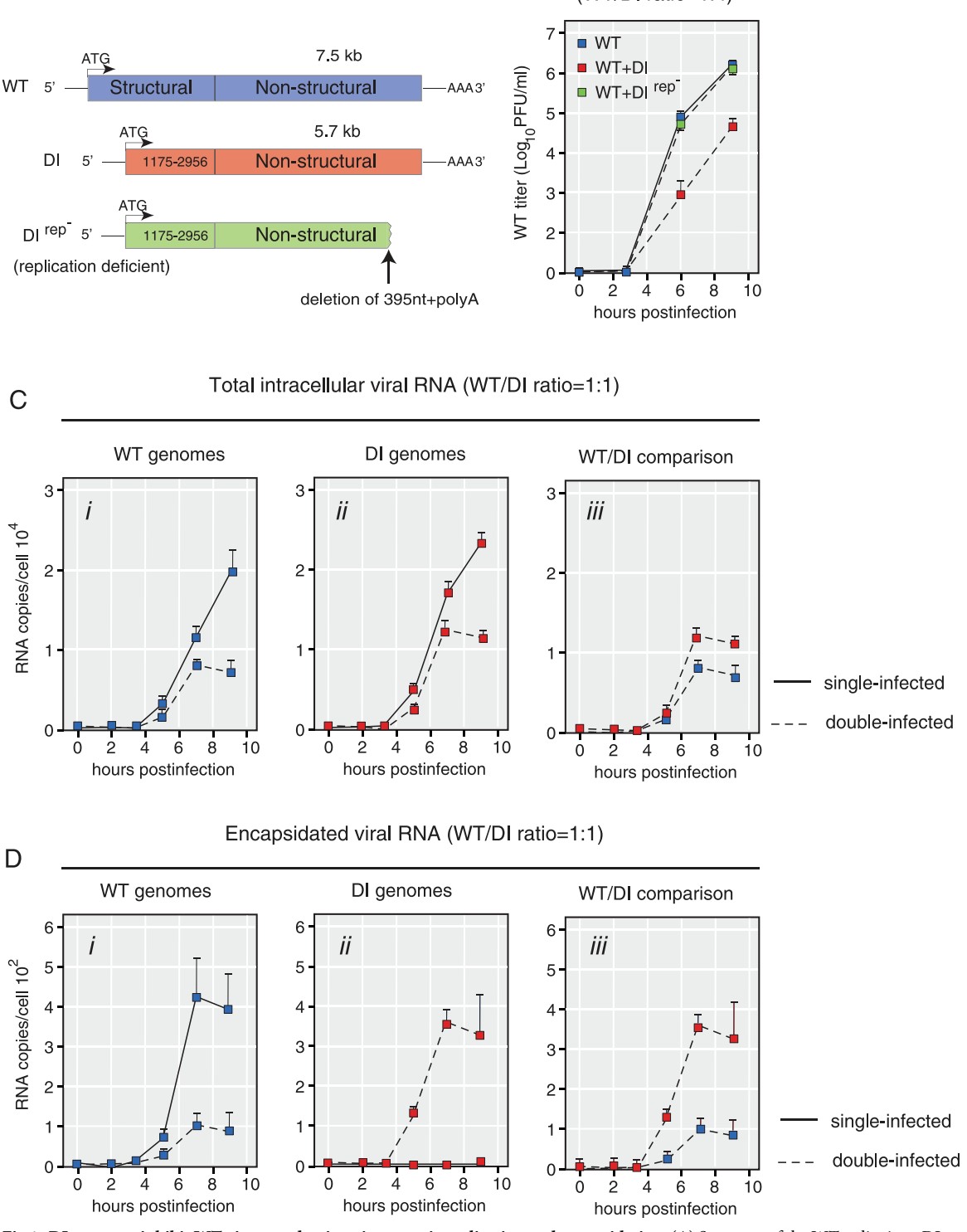

**Fig 1. DI genomes inhibit WT virus production via genomic replication and encapsidation.** (A) Structure of the WT poliovirus, DI (Δ1175–2956), and DI(Δ1175–2956)(Rep-). The DI genome has an in-frame deletion in its P1-encoding region. The PvuII-cut DI genome is used for a non-replicating RNA control (Rep-). (B) Growth curves of WT poliovirus after transfection of WT genomes and/or DI genomes (sampling time-points: 0, 3, 6 and 9 hours after transfection). (C, D) Copy numbers of total (C) or encapsidated (D) genomes over time after transfection (sampling time-points: 0, 2, 3.5, 5, 7 and 9 hours post transfection). Blue and red squares indicate

copy numbers of WT and DI genomes, respectively. Solid lines indicate single-transfected (WT or DI genome-transfected) samples, while dotted lines indicate double-transfected (both WT and DI genomes-transfected) samples. n = 3, mean ± standard deviations (SD) (B-D).

## Quantification of the copy number of WT and DI genomes following co-transfection

Next, we examined the interaction between DI and WT genomes by varying the ratio of each RNA used to initiate transfection. As DI RNA demonstrated an increase in transfection efficiency, probably due to its shorter genome length, we adjusted DI and WT RNA concentrations to achieve equal number of transfected cells. Given that DI genomes are $\sim 2,000$ nucleotide shorter ($\sim 1/4$ shorter) than WT genomes, it is expected to replicate faster than WT, and thus the copy number of DI genomes are higher than that of WT genomes. Transfection of shorter genomes is also more efficient than larger RNAs. We thus optimized our protocol to deliver equal copy number of DI and WT genomes into the transfected cells. We transfected 5$\mu$g of WT to 1.25$\mu$g of DI genomes, and we collected RNA samples at given time-points (t = 0, 2, 3.5, 5, 7 and 9 hours after transfection). The average number of genomes in a single cell was estimated as the total number of genomes divided by the number of transfected cells (successfully transfected cells were 25.7%). Replication of WT and DI decreased $\sim 7$ hours after co-transfection, but this effect was not observed in the cells transfected only with WT or DI (Fig 1C). Thus, replication of WT genomes was inhibited by DI, and the number of accumulated DI genomes also decreased in the presence of WT. This suggests that WT and DI genomes compete with each other for one or more limiting factors required for replication. Nonetheless, DI genomes have an small advantage and accumulated at slightly higher level than WT genomes (Fig 1C). To determine the numbers of encapsidated WT and DI genomes, we treated supernatants of infected cells with a mixture of RNase A and RNase T1. Viral RNAs encapsidated in virus particles are resistant to RNase treatment, while naked RNAs are degraded. Therefore, the number of RNase-resistant genomes is a measurement of particles containing WT and DI RNA. Of note, after the RNase treatment, DI genomes were not detected in samples obtained from cells transfected with DI RNA alone (Fig 1D*i*), indicating, as expected, that without trans-encapsidation by capsid proteins provided by WT DI RNA is sussceptible to RNase treatment. The decrease of encapsidated WT genomes between singly and co-transfected cells conditions was two-fold larger than that of WT genomes without RNase-treatment, consistent with the idea that DI genomes hamper WT genome encapsidation (Fig 1C and 1D, compare the difference between plain and dashed blue lines at 9 hours after transfection in Fig 1D*i* to the difference in Fig 1C*i*).

These results support the idea that DI RNAs replicate faster than WT genomes, due to their shorter genome, consistent with previous observations [33, 34]. Interestingly, co-transfection results in a reduction in replication of both WT and DI genomes most likely due to competition for some host-cell limiting factor needed for genome amplification. In addition, capsid proteins produced by WT genomes limit DI and WT virus production as DI genomes compete for these proteins and thus further inhibit WT production. To further examine the mechanism of defective interference and quantitatively evaluate the effect of co-replicating DIs, we designed a simple mathematical model that describes the DI/WT genome interactions.

## Mathematical description of intracellular competition

A deterministic mathematical model describing the intracellular competition between DI and WT genomes was developed, adapted from an existing competition model for human immunodeficiency virus (HIV) [35]. In order to describe appropriately the intracellular dynamics of poliovirus, we explicitly account for limiting resources depleted by the virus during replication, slowing down the growth of the population over time. This slowdown was experimentally shown by [36], who reported an exponential growth of viral RNA up to the third hour of infection, followed by a linear increase and then a plateau. The effect of limiting resources on poliovirus replication has been investigated by [37] using a mathematical model to explain the observed saturation in viral replication dynamics. Resources depleted by the virus may include phospholipids and *de novo* synthesized membranes for the formation of replication organelles [38, 39], pro-virus host factors required for virus translation, replication or encapsidation [40], or even the supply of amino acids for protein synthesis or nucleotides for genome replication [37]. Our model considers a generic set of resources ($R$) at the virus disposal during replication. It describes the changes, over the course of infection of a cell, of the numbers of WT ($G_{WT}$) and DI ($G_{DI}$) RNA genome copies, of free capsids produced by the WT ($C$) and of limiting resource units ($R$) depleted by the genomes for their replication. The set of ODEs is the following:

$$\frac{dG_{WT}}{dt} = \theta\varepsilon G_{WT}R - c_g\kappa CG_{WT} - \alpha G_{WT} \tag{1}$$

$$\frac{dC}{dt} = \eta\theta\varepsilon G_{WT}R - \kappa(G_{WT} + \omega G_{DI})C - \beta C \tag{2}$$

$$\frac{dG_{DI}}{dt} = P\theta\varepsilon G_{DI}R - \omega c_g\kappa CG_{DI} - \alpha G_{DI} \tag{3}$$

$$\frac{dR}{dt} = \lambda - c_r\varepsilon(G_{WT} + G_{DI})R - \gamma R \qquad R(0) = \lambda/\gamma \tag{4}$$

Model parameters are summarized in Table 1 and can be described through three distinct stages of the viral cycle: replication, capsid synthesis and encapsidation. A flow diagram of the model is available in Fig 2. Limiting resources are produced at a constant linear rate $\lambda$ and captured by WT ($G_{WT}$) and DI ($G_{DI}$) genomes at a rate $\varepsilon$ per unit of resource (uor) per minute. One uor and one viral genome, by definition, form one replication complex ($c_r$ = 1 uor · genome$^{-1}$, [41]). Conditionally on the capture of a resource unit, a WT genome replicates and turns into $\theta$ genomes, before the replication complex disintegrates (we set the condition $\theta > 1$ in order for virus genomes to replicate). We assume that this happens quickly compared to the other processes. The replication rate of WT genomes is given by the product $\theta\varepsilon R(t)$. The replication rate of DI genomes is faster than that of WT genomes by a fixed factor $P$ ($P > 1$), which likely depends of their shorter genome size requiring less time for the polymerase to complete a round of template copying [33, 34]. Because DI genomes lack the genes responsible for capsid proteins synthesis, only WT genomes are capable of producing free capsids ($C$), with the capsid-to-genome accumulation ratio $\eta$. WT genomes are then encapsidated (i.e. packaged into free capsids) at rate $\kappa$. DI genomes are set to encapsidate faster by a fixed factor $\omega$ ($\omega > 1$), based on experimental observations from Fig 1C and 1D. One viral genome encapsidates into one capsid to form a virion ($c_g$ = 1 genome · capsid$^{-1}$). Finally, $\alpha$, $\beta$, and $\gamma$ represent the decay rates of, respectively, viral genomes, free capsids and resources.

**Table 1. Notations used in the model and model parameters.**

| Notation | Definition | Unit [a] | | | |
|---|---|---|---|---|---|
| | **Observed variables** | | | | |
| $g_{WT}^{tot}, g_{DI}^{tot}$ | WT, DI total genome number | gen | | | |
| $v_{WT}, v_{DI}$ | WT, DI virion number | gen | | | |
| | **State variables** | | | | |
| $G_{WT}, G_{DI}$ | WT, DI genome number | gen | | | |
| $C$ | Free capsid number | caps | | | |
| $R$ | Resource units | uor | | | |
| $C_{WT}, C_{DI}$ | WT, DI virion number | vir | | | |
| | **Model parameters** | | **Best value [c]** | **Confidence Interval [d]** | **Sensitivity analysis [e]** |
| $c_g$ | Number of viral genomes per capsid | $gen \cdot caps^{-1}$ | 1[b] | | |
| $c_r$ | Number of uor per viral genome | $uor \cdot gen^{-1}$ | 1[b] | | |
| $\theta$ | Genome replication factor | - | 1.121 | [1.000; 26.983] | [1, 1.25, 1.5, 1.75, 2] |
| $\varepsilon$ | Resource capture rate by viral genomes | $(uor \cdot min)^{-1}$ | $2.021 \cdot 10^{-6}$ | $[1.355; 3.000] \times 10^{-6}$ | $[0.8, 1.3, 1.8, 2.3, 2.8] \times 10^{-6}$ |
| $\kappa$ | Encapsidation rate | $(gen \cdot min)^{-1}$ | $5.097 \cdot 10^{-6}$ | $[4.900; 5.282] \times 10^{-6}$ | $[2, 3.5, 5, 6.5, 8] \times 10^{-6}$ |
| $\alpha$ | Viral genome decay rate | $min^{-1}$ | 0 | - | 0 |
| $\eta$ | Capsid to genome accumulation ratio | $caps \cdot gen^{-1}$ | $5.260 \cdot 10^{-2}$ | $[5.058; 5.439] \times 10^{-2}$ | $[2, 3.5, 5, 6.5, 8] \times 10^{-2}$ |
| $\beta$ | Capsid decay rate | $min^{-1}$ | $2.589 \cdot 10^{-3}$ | $[1.659; 3.476] \times 10^{-3}$ | $[1, 1.75, 2.5, 3.25, 4] \times 10^{-3}$ |
| $P$ | DI-to-WT replication ratio | - | 1.075 | [1.062; 1.087] | [1, 1.15, 1.3, 1.45, 1.6] |
| $\omega$ | DI-to-WT encapsidation ratio | - | 2.185 | [1.943; 2.453] | [1, 1.75, 2.5, 3.25, 4] |
| $\lambda$ | Resource production rate | $uor \cdot min^{-1}$ | 10.008 | [0.497; 88.924] | $[1, 1.25, 1.5, 1.75, 2] \times 10^{-3}$ |
| $\gamma$ | Resource decay rate | $min^{-1}$ | $7.471 \cdot 10^{-4}$ | $[7.426; 88.582] \times 10^{-4}$ | $[3.5, 5.25, 7, 8.75, 10.5] \times 10^{-8}$ |
| $L$ | Logistic's maximum | $min^{-1}$ | $3.078 \cdot 10^{-2}$ | $[3.061; 3.099] \times 10^{-2}$ | - |
| $s$ | Logistic's steepness | $min^{-1}$ | $3.234 \cdot 10^{-2}$ | $[3.234; 3.234] \times 10^{-2}$ | - |
| $t_0$ | Logistic's midpoint | min | 318.459 | [318.459; 318.459] | - |

[a] gen: genomes; caps: capsids; uor: units of limiting resource; vir: virions; '-': dimensionless

[b] Fixed parameter values

[c] Best optimized value shared by the 85 first sets (identical until fourth significant digits) for reduced model optimization ($P$, $\omega$, $\kappa$, $\eta$, $\alpha$, $\beta$, $L$, $s$, $t_0$), and arbitrarily chosen best set for full model optimization ($\varepsilon$, $\theta$, $\lambda$, $\gamma$). These best values are used in Fig 3 and S2 Fig, and S1 Table (model $\mathcal{M}^{123}$).

[d] Range of variation over 150 best sets of parameter values

[e] The set of best values estimated from the full model and taken for the sensitivity analysis and all of the theoretical modelling work (i.e. Figs 4, 5 and 6, S3 and S5 Figs) was: $\theta = 1.192$, $\varepsilon = 1.814 \cdot 10^{-6} (uor \cdot min)^{-1}$, $\lambda = 1.000 \cdot 10^{-3} uor \cdot min^{-1}$ and $\gamma = 7.157 \cdot 10^{-8} min^{-1}$.

The number of encapsidated WT genomes (i.e. WT virions, $C_{WT}$) and DI genomes (i.e. DI virions, $C_{DI}$) were measured experimentally (Fig 1D) and can easily be derived from Eq 2 as the loss of free capsids due to encapsidation:

$$\frac{dC_{WT}}{dt} = \kappa C G_{WT} \tag{5}$$

$$\frac{dC_{DI}}{dt} = \omega \kappa C G_{DI} \tag{6}$$

Further, burst sizes, which are defined as the number of virions at cell lysis, i.e. 9 hours post transfection (hpt, [42, 43]), can be written as $\mathcal{B}_{WT} = C_{WT}(9\,hpt)$ for WT virions and $\mathcal{B}_{DI} = C_{DI}(9\,hpt)$ for DI virions.

The system of Eqs (1)–(6) encompasses 10 parameters and six variables, among which only four variables could be experimentally measured. Therefore, the mathematical model presents

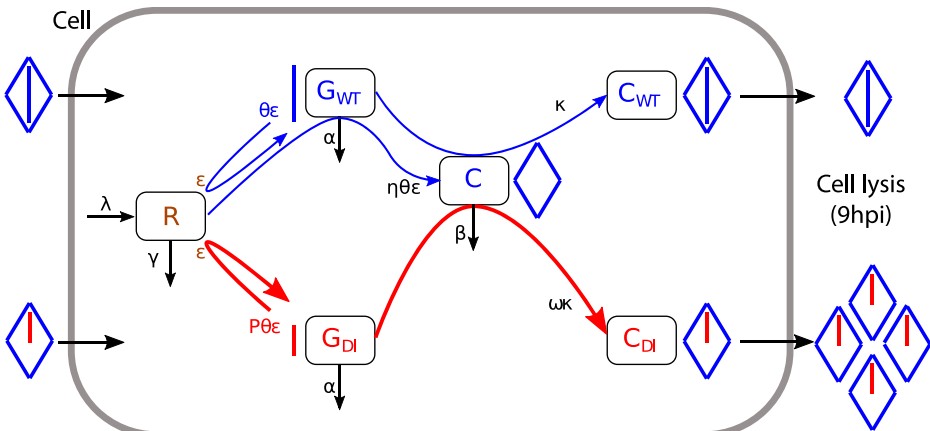

**Fig 2. Flow diagram of the model (Eqs (1)–(6)).** State variables: resource units number $R$; wild-type (WT) poliovirus naked genome number $G_{WT}$; defective interfering (DI) naked genome number $G_{DI}$; WT-produced capsid proteins number $C$; encapsidated WT genome number $C_{WT}$; encapsidated DI genome number $C_{WT}$. Model parameters are defined in Table 1. Color code is blue for WT, red for DI and brown for resources. Segments represent genomes and diamonds represent capids.

a classical problem of parameter identifiability, specifically regarding parameters $\varepsilon$ and $\theta$ that appear as a product, with parameter $\varepsilon$ figuring separately in Eq (4) corresponding to the unmeasured variable $R$. To solve this problem, we built a reduced model by assuming that the decrease in resources due to viral uptake follows a logistic decreasing function. We use a two-step optimization procedure to estimate the parameters. Briefly, we first estimate a set of parameters using the reduced model. Then, we use the parameters that are common between the reduced and the full models as "best guess" and run the estimation on the full model. Using this procedure, we are able to estimate the model parameters with reasonably high confidence (see Material and methods and Table 1).

We additionally conducted a model selection procedure in order to validate that the features introduced in our model were improving fitting statistics chosen as (i) the R-squared between experimental and fitted data, and (ii) the log-likelihood and (iii) Akaike information criterion (AIC) of a linear model explaining the experimental data with the fitted data. The different versions of the model that were tested are detailed in S1 Text and the results are presented in S1 Table. In what follows we will present results from both the reduced and full models for comparison. We report a full comparison of the models in the S1 Text.

Model predictions fit well the experimental measurements in Fig 1C and 1D, with best $R^2$ = 0.974 for the reduced model and 0.964 for the full model (Fig 3). In dually transfected cells, both versions of the model reproduce well the number of naked and encapsidated genome copies. In singly transfected cells, the reduced model somewhat underestimates the number of naked genome copies while fitting satisfactorily the number of encapsidated genome copies. Conversely, the full model describes well the number of naked genome copies but overestimates the number of encapsidated genome copies. Overall, either model is able to describe the reduction on genome replication when WT and DI are co-transfected as compared to singly transfected cells (compare same color plain or dashed curves between Fig 3A and 3B). We hypothesized that this effect is the consequence of competing for limiting resources necessary for replication. The model also shows that the WT is more severally affected by competition for resources than the DI, as observed experimentally (Fig 3A, compare red and blue plain curves) thanks to the higher replication rate of DI genomes ($P$ = 1.075, Table 1). Model predictions are also consistent with the experimental observation that DI genomes are more

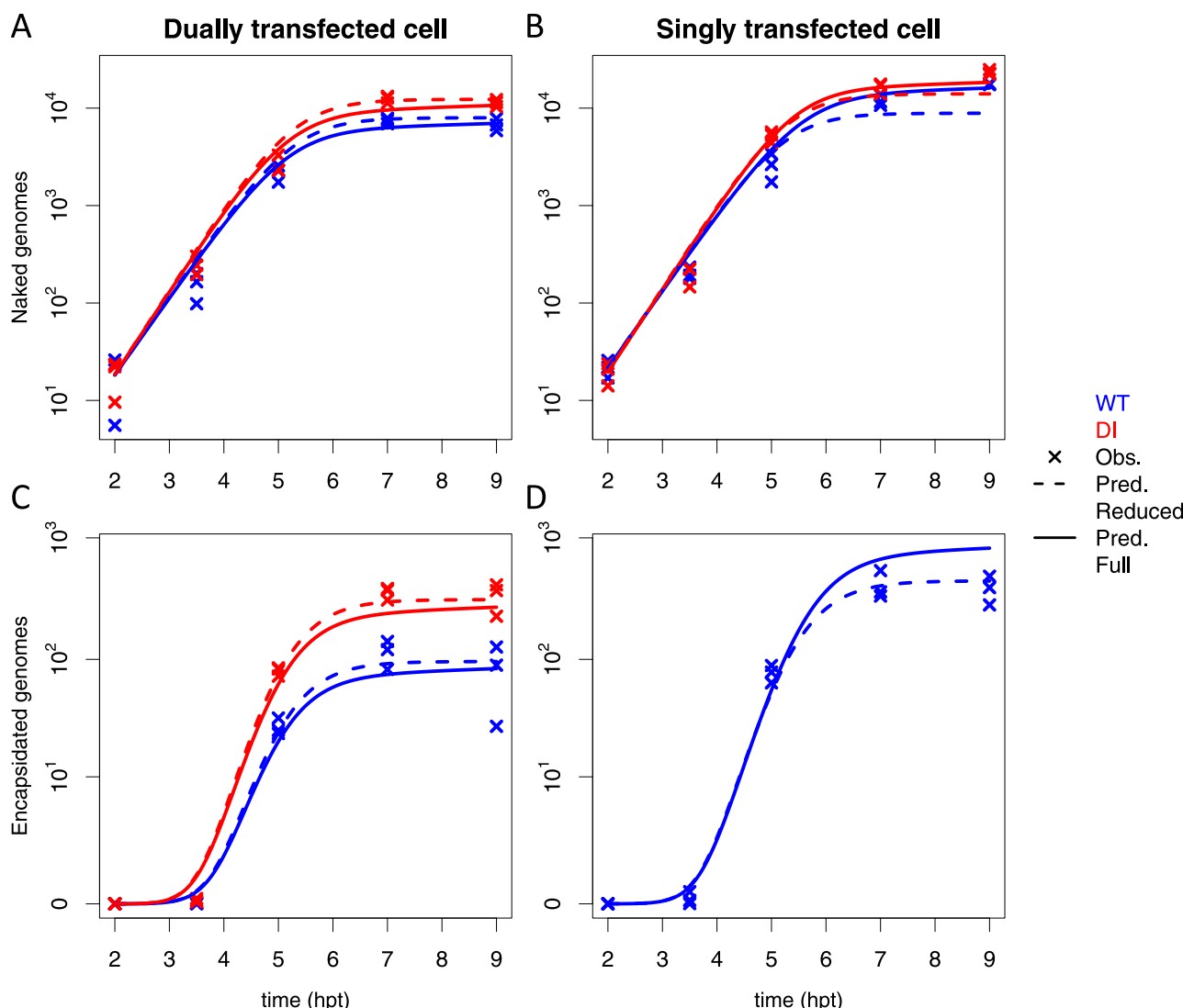

**Fig 3. Experimental data and model fit for the naked and encapsidated wild-type (WT) and defective interfering (DI) genomes.** Evolution of the number of WT and DI (A-B) naked genome copies and (C-D) encapsidated genome copies with time, from 2 to 9 hours post transfection (hpt). (A & C) show data in dually infected cells whereas (B & D) show data in singly infected cells. WT and DI results are shown in blue and red color, respectively. Crosses indicate experimental data for 3 replicates per sampling time at 2, 3.5, 5, 7 and 9 hpt (data drawn from Fig 1C and 1D, see Material and methods for details). Dashed curves represent the fit of the reduced model with logistic equation and solid curves show the fit of the full model (Eqs (1)–(6)).

efficiently encapsidated than WT genomic RNA (Fig 3C, compare red and blue plain curves). This is due to the higher encapsidation rate of DI genomes ($\omega = 2.185$, Table 1). Indeed, if we introduced in the model the same encapsidation rate for WT and DI genomes (i.e. $\omega = 1$, model $\mathcal{M}^{12}$ in S1 Text, eqs. (S10)–(S13)) we observed an underestimation of DI encapsidated genomes as compare with the experimental measurements. We conclude that DI genomes are encapsidated more efficiently than WT RNA. It is possible that the faster RNA replication of DI genomes provide an advantage on the competition for capsid proteins during encapsidation (S1 Fig). Thus, the model describes the experimental observation that WT genomes are encapsidated at a decreased rate in dual transfection compared to single WT transfections (compare Fig 3C and 3D, blue curves).

All parameter estimates are narrowly defined by the fitting procedure (Table 1) except for $\theta$, $\lambda$ and $\gamma$ as they tend to correlate with each other, yielding non-uniqueness of best-fit parameter values (S2(A), S2(C) and S2(D) Fig, see also the distribution of $\varepsilon$ values in S2B Fig). Also, a strong log-to-log relationship was found between the resource production to decay ratio $\lambda/\gamma$ and the replication factor $\theta$ (S2(E) Fig).

Both the reduced and full models feature a time-dependent virus replication rate (S2(F) Fig). In the reduced model, this is given by the logistic function $\Lambda(t)$ (Eq 10 in Material and Methods), and in the full model by the product $\theta\varepsilon R(t)$. In both models, the best fit yields approximately the same time-dependent replication rate, starting at $3.07 \cdot 10^{-2}\,\mathrm{min}^{-1}$ for the reduced model and at $3.04 \cdot 10^{-2}\,\mathrm{min}^{-1}$ ($\theta\varepsilon\lambda/\gamma$) for the full model, and decreasing with time towards 0. According to the reduced model, the time of half-decay is 318 minutes ($t_0$ in Table 1), which corresponds to 5.3 hours post transfection.

The predictive power of the full model with best estimated parameter values was tested on independent experimental measurements of WT burst size corresponding to various initial DI-to-WT ratios for which the model had not been trained (S3 Fig). Relative experimental and predicted WT burst sizes were normalized by their respective value for WT-only transfection. The model was able to predict experimental outputs well, albeit underestimating WT output for some DI-to-WT input ratios. The largest underestimation was observed for the DI-to-WT input ratio of 0.5, and this discrepancy vanished as the input ratio increased.

## Model predictions

The aim of our work is to understand the competition dynamics of WT and DI genomes during co-infection. To achieve this goal, we used the model described above to study how changes in parameter values around their experimental estimates can impact the outcome of the competition. Additionally, we also used the model to evaluate the effect of initial infection conditions, such as initial genome copy numbers of WT and DI and a time delay of cell infection on their respective burst sizes.

**Sensitivity analysis.** A sensitivity analysis was performed to identify parameters that have a significant effect on the output variable of interest, which we set as the proportion of WT virions at the time of cell lysis, $\Phi_{WT}$ (Eq 12). Parameters were varied by $\pm50\%$ of a set of best fit value based on experimental data, with five equally spaced values for each parameter (Table 1). The decay rate of genomes was not varied, because it was estimated to be negligible. The results indicate that the DI-to-WT replication ratio, $P$, and the DI-to-WT encapsidation ratio, $\omega$, as well as their second-order interaction, were the most influential factors for the variation of $\Phi_{WT}$, explaining 76%, 17% and 7% of the variance, respectively (Fig 4A). All the remaining factors and their second-order interactions had a negligible effect (less than 1% of the variance). Hence, our model predicts that only parameters associated with the DI design have a strong impact on the degree of suppression of WT by DI.

To further examine the effect of $P$ and $\omega$, we varied both parameters from their best estimated value (Fig 4B). As expected from the global sensitivity analysis, $P$ was found to be more important than $\omega$ for the production of WT virions, the gradient of $\Phi_{WT}$ being steeper along $P$-axis than along $\omega$-axis. Within the tested range of parameters $P$ and $\omega$, the value of $\Phi_{WT}$ varied between 2% and 50%. The reference value of $\Phi_{WT}$ corresponding to best-fit parameter estimates from experimental data was 23%. Therefore, we can predict that a DI particle with a lower replication rate, or, to a lesser extent, with a lower encapsidation rate than the DI particle used in the present work would weaken its competitivity with the WT virus, potentially leading to an increase in the proportion of WT virions at cell lysis of up to 27%. Conversely, a DI

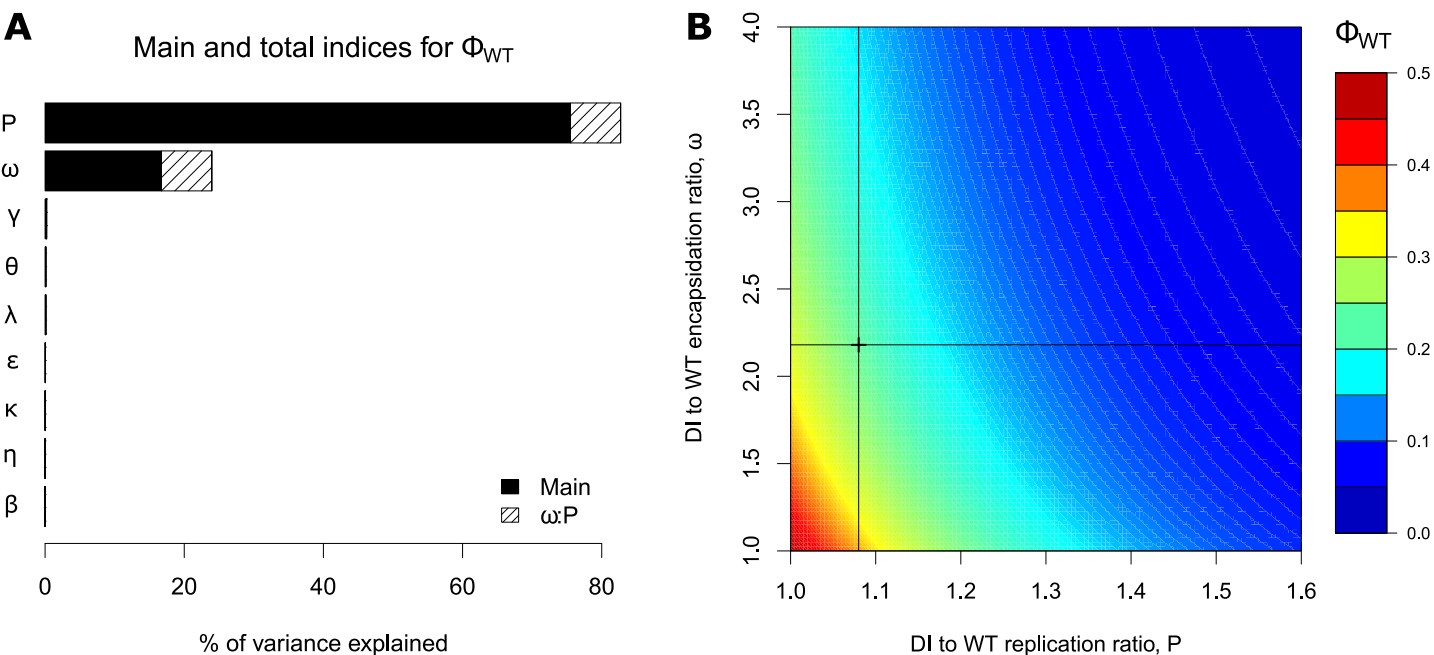

**Fig 4. Sensitivity indices for the proportion of WT virions at cell lysis ($\Phi_{WT}$) and the impact of the most important parameters.** A: Main, total and most important interaction indices for $\Phi_{WT}$. Main indices correspond to single parameter effect (black parts), and total indices add the effect of the factor in interaction with all other factors (second-order interactions, full bars). Hatched areas represent the importance of the strongest pairwise interaction between defective interfering (DI) to WT encapsidation ratio $\omega$ and DI to WT replication ratio $P$. B: Heat map representing the impact of $P$ and $\omega$ on $\Phi_{WT}$, all other parameters being fixed to their best estimated value. The cross and lines in (B) show estimated parameter values from fitting the model to the experimental data.

particle characterized by a higher replication rate or a higher encapsidation rate would strengthen its competitivity, potentially leading to a decrease in $\Phi_{WT}$ of up to 21%.

**Validation of model predictions using DI variants.** To evaluate predictions of the model, we next investigated the competition between WT virus and DI variants. Our sensitivity analysis provides a quantitative measurement of the WT/DI competition as a function of the replication advantage of the DIs (DI-to-WT replication ratio $P$). As assessed in the previous Section, the most significant determinant for the outcome of infection is the ratio between DI and WT replication, $P$. Moreover, the simplest model that explains the difference between these mutants and the WT assumes no difference in encapsidation efficiency. The only nostructural protein involved in encapsidation is 2C, and none of the mutations we consider occur in that protein. To validate these predictions we isolated DI variants with defined replication advantage. In this experiment, we used WT poliovirus type 3 (PV3) and PV1-DI to discriminate their genomes by sequencing. Briefly, PV1-DI and WT PV3 were serially passaged for 8 times (Fig 5A and S1 Text). We determined PV3 and DIP titers at each passage to gain information on the dynamics of the co-infection experiments (Fig 5B). Both DI and PV3 compete with each other, resulting in a 10-folds reduction on PV3 titer, and the establishment of an equilibrium in which both PV3 and DIs are maintained for 8 passages (Fig 5B, red and green squares).

These experiments also serve to determine whether mutations accumulate during co-passaging in either DI or PV3 genomes. This information was used to construct DI particles with defined replication advantage with respect to the original DI (DI Ori). No change in encapsidation efficiency is expected. The mutation rates in RNA viruses can range from $10^{-4}$ to $10^{-6}$ per base, while conventional next-generation sequencing (NGS) can only detect variants at a

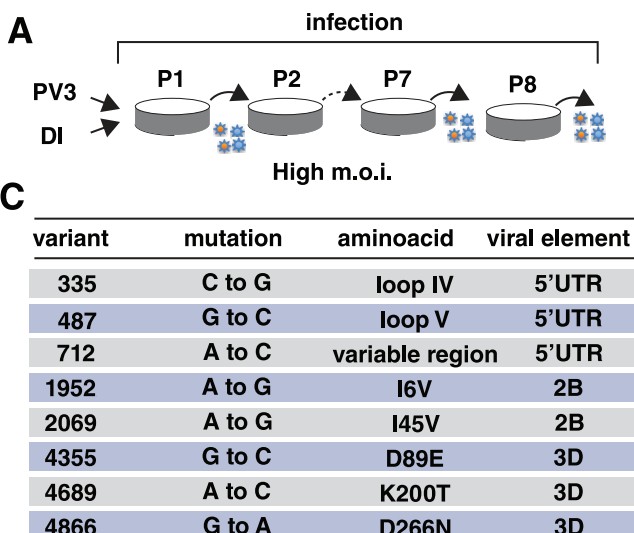

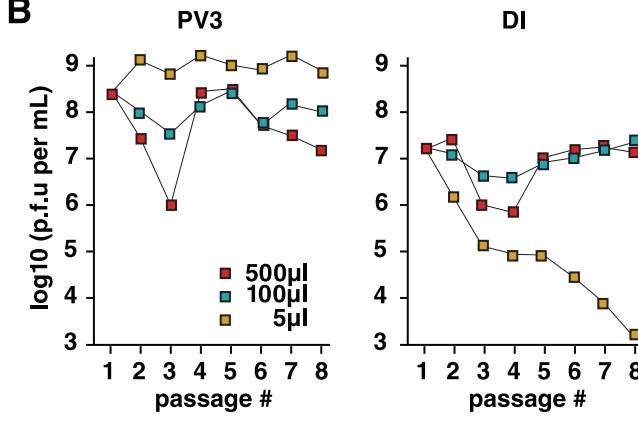

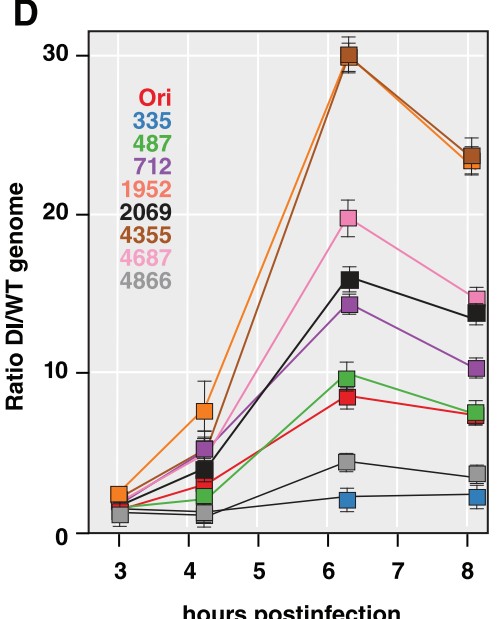

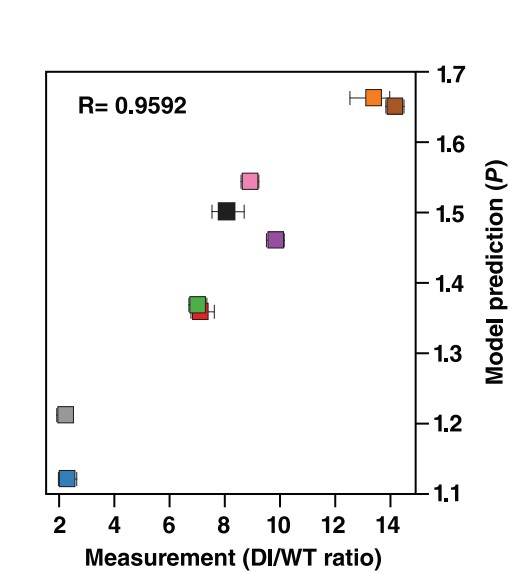

**Fig 5. Investigating DI mutant genomes.** A: PV3 and PV1-DI particles were serially passaged 8 times. B: Titers of PV3 (PFU/mL) and PV1-DI particles (IU/mL) included in the samples were determined. C: Mutations in the DI genomes positively selected over passages. D: One-step replication cycle. Dynamics of the ratio of PV1-DI to (wild-type) WT PV1 naked genome copies (total RNA) over time (3, 4.33, 6.33 and 8.17 hours post transfection), for distinct DI mutants (Ori: original non-mutant DI genome). E: Estimation of the ratio of PV1-DI to WT PV1 replication rate *P* from a least square regression between experimental and model predicted PV1-DI to WT PV1 naked genome ratios. Confidence intervals represent values of *P* at 10% of the minimum least square value. Model predictions of *P* against experimental DI to WT naked genome ratios at the last sampling time-point.

frequency of about 1 in 100–500 (due to sequencing errors). Consequently, many variants in a viral population which exist at low frequencies cannot be distinguished from noise using conventional NGS. Therefore, we used Circular Sequencing (CirSeq), a technique developed to improve accuracy of NGS [44, 45]. We engineered several of the mutations identified into the DI cDNA, which facilitates identification of rare variants detection in the population. We identified eight mutations positively selected over passages (Fig 5C).

Next, we engineered DI variants carrying the identified mutations and determined the outcome of competition between WT virus and DI variants. HeLaS3 cells were co-transfected with WT poliovirus and DI variants and the ratio of DI-to-WT genome copies was estimated by RT-PCR over the course of infection (Fig 5D and S1 Text). In this way we were able to estimate genome replication $P$ using the model, with all other parameters fixed to a set of estimated best values from the main experiment (Table 1 and S1 Text), and we established that the model prediction $P$ and the experimental determined WT/DI ratios correlate with a Pearson coefficient of 0.96 (Fig 5E). Thus, these data confirms that, as predicted by our sensitivity analysis, the relative DI/WT replication ratios predict the outcome of infection. Thus our model serves as a useful tool to identify the most important determinants on the DI/WT interaction and enable to quantitatively estimate key parameters controlling modulation of the WT infection.

**Effect of the multiplicity of infection and the timing of co-infection on WT burst size.**
We use our model to investigate the impact of varying initial conditions including (i) the time difference between WT and DI infection of the cell and (ii) the initial quantities of WT and DI on the WT and DI burst sizes, $\mathcal{B}_{WT}$ and $\mathcal{B}_{DI}$, respectively. First, the time for DI infection compared to WT was varied from -7 to +7 hours post WT virus infection (Fig 6A). Negative delay values indicate that DI infects first, while positive values indicate that WT infects first. WT was allowed to produce at least one virion at cell lysis when DI was infecting the cell no more than 1.37 hours prior to the WT virus (delay = −1.37 hours). From this delay value, WT burst size increased as a steep logistic function, reaching a plateau at $\mathcal{B}_{WT} = 811$ virions from a delay for DI infection of around 4 hours post WT infection. The curve of DI burst size was bell-shaped, reaching a maximum of $\mathcal{B}_{DI} = 298$ virions at a delay for DI infection of 0.4 hours post WT infection. The delay window allowing DI genome to be encapsided and produce at least 1 virion is narrow, from -3.43 to 4.37 hours. Most importantly, DI burst size superseded WT burst size only until the delay of 0.62 hours. The difference between DI and WT burst sizes was the largest when DI infected the cell 0.03 hours prior to WT ($\mathcal{B}_{DI} - \mathcal{B}_{WT} = 181$ virions).

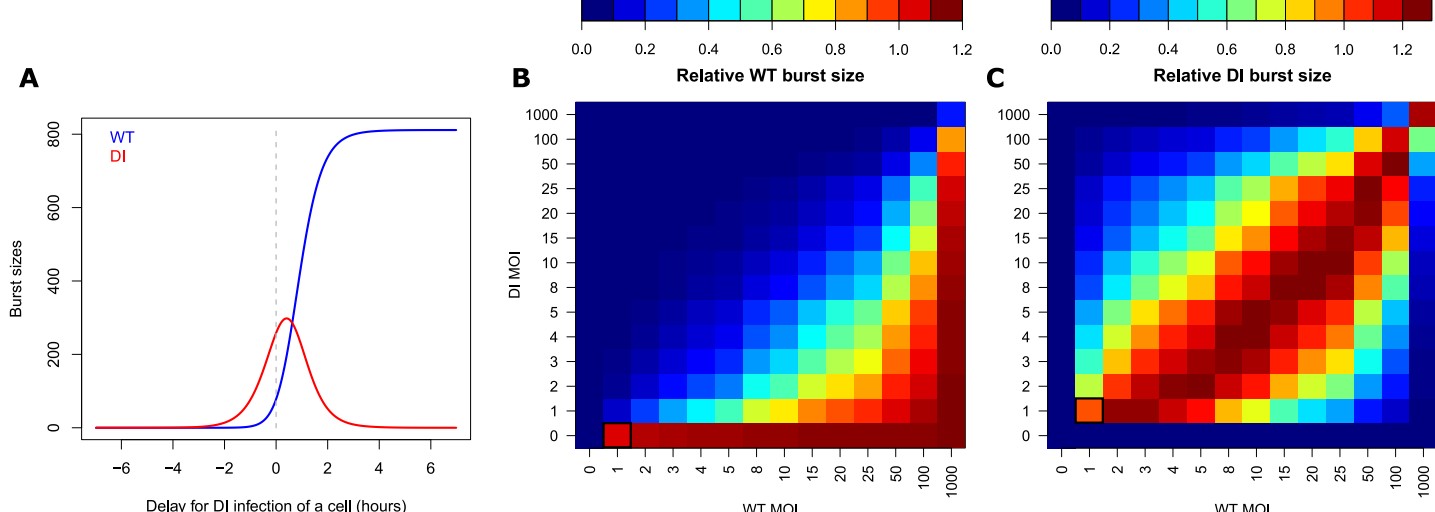

**Fig 6. Model-predicted impact of infection delay and initial multiplicity of infection (MOI) on WT and DI burst sizes.** A: WT (blue) and DI (red) predicted burst sizes (encapsided genomes at 9 hours post infection) for various delays in DI infection of a cell. Negative delays correspond to cases where DI infects first and positive delays to cases where WT infects first. The dashed gray line marks the case of simultaneous infection by WT and DI. B-C: Heat maps representing the predicted relative burst sizes of (B) WT, $\mathcal{B}_{WT}$, and (C) DI, $\mathcal{B}_{DI}$, as a function of initial WT (x-axis) and DI (y-axis) MOIs (input). WT burst sizes were normalized by WT burst size resulting from WT:DI = 1:0 MOIs and DI burst sizes by DI burst size resulting from WT:DI = 1:1 MOIs (black squares).

We then investigated the impact of varying initial WT and DI multiplicities of infection (MOIs, corresponding to the number of viral genomes successfully entering a cell and initiating infection) from 0 to 1000 on WT and DI relative burst sizes (Fig 6B and 6C). WT burst sizes were normalized by WT burst size obtained for WT:DI = 1:0 MOIs, while DI burst sizes were normalized by DI burst size obtained for WT:DI = 1:1 MOIs. WT relative burst size varied from 0 to 1.11 depending on MOIs. DI relative burst size ranged from 0 to 1.26.

For WT virions to be completely suppressed, DI must infect the cell 2 hours prior to WT infection or at higher MOI than the WT, complete suppression taking place at co-infection of 100 DI and 1 WT. Conversely, the optimal initial conditions for maximizing the DI burst size arrive when WT is initially present in a slightly larger quantity than DI. The DI needs enough WT to exploit its capsids and produce virions. Because (i) the DI replicates and encapsidates faster than the WT, (ii) only the WT produces free capsids and (iii) replication and capsid production result in resource depletion, it is better for DI virion production to have the WT infect a cell in a slightly higher quantity than the DI. In that case, the WT has a slight initial advantage over the DI and can use resources to produce free capsids. In return, the DI can exploit those free capsids at its own advantage as it replicates and encapsidates more efficiently.

We also examined the cross-effect of the time delay and the variation of initial MOIs (S5(A) Fig). Globally, the shapes of WT and DI burst size curves as a function of delay are very similar between different WT and DI MOIs. The observed effect is a shift of the curves on the delay axis. Equal WT and DI MOIs generate very similar WT and DI burst size curves as a function of delay time. At these equal MOIs, DI competes more efficiently than WT upon co-infection, i.e. simultaneous infection of the cell by DI and WT (S5(A) and S5(B) Fig) and maximal DI burst size occurs when DI infects the cell around 0.3–0.5 hours after the WT (S5(A) Fig). As WT MOI gets larger than DI MOI, both the delay time that maximizes the DI to WT burst size difference and the peak of DI burst size shift towards delays where DI infects the cell before the WT (S5(A), S5(B) and S5(C) Fig). Giving an initial MOI advantage to the WT results in the DI having to infect the cell sooner in order to maximize production. On the opposite, as DI MOI gets larger than WT MOI, the shift is towards delays where WT infects the cell before the DI. Giving the DI an additional advantage in terms of MOI than it already has by being faster at replicating and encapsidating can be harmful for DI production, but this can be compensated by an earlier infection of the WT.

## Discussion

We used a combination of mathematical modeling and empirical measurements to better understand of the interaction between a cooperator, the WT, producing capsid proteins as public goods, and a cheater, the DI, exploiting those capsid proteins from the WT. This type of direct competition affecting the growth of a competitor represents a case of interference competition. In addition, in co-infected cells, we also observed that WT and DI compete for shared resources necessary for replication, a phenomenon known as exploitation competition [1–3]. Hence our study identifies two different types of competition occurring during the co-infection of a cell by WT and DI.

Huang & Baltimore [46] argue that defective particles may influence the outcome viral infections. DI RNAs are produced by RNA replication errors, and are maintain in the virus population by high-multiplicity of infection, where DI and WT virus co-infected cells [47]. Internal sequences of the original vRNA of DI RNA segments are deleted, but cis-acting elements are retained, which enables DI replication. Naturally occurring immunostimulatory defective viral genomes (iDVGs) are generated during respiratory syncytial virus (RSV)

replication and are strong inducers of the innate/natural antiviral immune response to RSV in mice and humans [25].

## Mechanism of defective interference

Our study provides evidence for three main conclusions. First, the number of WT or DI genomes, taken individually, is lower in dually infected cells compared to singly infected cells (Fig 1C*i* and 1C*ii*), indicating that they compete for a limiting resource for replication. Second, in dually infected cells, DI genomes replicate faster than WT genomes (Fig 1C*iii*), showing the advantage of their shorter genome size [33, 34]. Furthermore, we identified DI variants with increase replication fitness to confirm the model prediction indicating that the WT/DI replication ratio is a major determinant for the outcome of infection. Third, our study also demonstrated WT genomes encapsidation is inhibited by co-infecting DIs (Fig 1D*i* and 1C*i*), which further decrease WT virions production.

We have designed a minimal mathematical model able to capture key features of the DI/WT interaction during a single-cell replication cycle. We accounted explicitly for depletion of cellular resources and available capsid proteins produced by the WT virus. The model accurately describes the experimental data, and to predict new data on which the model had not been trained (S3 Fig). Fitting the available experimental data enables estimating parameters within biologically realistic ranges providing a better how the interaction of these processes affect WT/DI interaction during co-infection.

If DI's replicate faster than WT simply because its shorter genome, the ratio of WT to DI genome replication should correspond to the genome lengths, that is 7515bp/5733bp = 1.311. However, the best optimized value of the corresponding parameter *P* was lower (1.075, Table 1). This underlines the complexity of the replication process and the value to develop reliable model to examine these type of processes. For example, poliovirus replication is stepwise process in which formation of replication complexes, follow for synthesis of negative- and positive-stranded viral RNAs may affect the overall kinetics of the process [48]. Thus, each of these steps might lower the average difference in replication rate between the WT and DI genomes over the course of infection of a cell. Indeed, when our model optimization procedure to the reduced model was performed keeping P fixed to the ratio in genome lengths, we observed a worse fit to the data (S6 Fig).

Our analysis suggests that some limiting resources are required and depleted during replication of viral genomes. Those resources might be lipids recruited by the viral machinery for the formation of replication complexes [39, 40] or host proteins involved in in different steps of viral replication [37]. When both WT and DI co-infect a cell, they are in competition for the exploitation of those shared and limiting resources. As DI replicates faster, it depletes resources faster than WT, affecting WT replication compared to WT-only infections [3]. Additionally, virus replication may represent a disruption for the cell physiology, which may lead depletion of resources at later stages of the replication cycle [37]. Consistent with this idea, fitting the models to the experimental data we estimated an initial replication rate of $3.04 \cdot 10^{-2} - 3.07 \cdot 10^{-2} \text{min}^{-1}$, representing initial replication steps where the resources are not limiting. As resources are depleted, the model predicts that replication rate will tend to 0 following a logistic function (S2(F) Fig).

## Insights on model fit to experimental data

The reduced model underestimates the number of viral genome copies in singly infected cells. The logistic equation considers the time-dependent genomic replication to be the same in both dually and singly infected cells. Under this assumption depletion of resources required

for replication is identical in singly or dually-infected cells. As a result, genome copies are well predicted in dually-infected cells but underestimated in singly-infected cells, given that resource consumption is lower in single-infected cells. In contrast, the full model is able to recapitulate the impact of resource depletion on RNA genome production in both dually and singly infected cells, as a specific variable for resources and mass action terms are considered within the full model.

However, the full model overestimates the number of encapsidated WT genomes in singly infected cells. This result suggests that the WT virus encapsidates less efficiently when it is alone than in DI co-infected cells. Our model assumes that the encapsidation rate of WT is the same in singly- and dually-infected cells ($\kappa$). However, it is possible that DI could facilitate WT encapsidation. Indeed, during co-infection, two viruses can exploit a common pool of resources equally [49]. Given that the WT and DI genomes encode non-structural proteins that involved in capsid processing (3C/3CD) or encapasidation (2C), co-infection may generate an excess of these viral factors increasing the effiency of WT encapsidation [50, 51].

Our experimental observations indicate that the number of encapsidated genomes decrease on average from 7 to 9 hours post transfection (from 410 to 385 for WT in singly infected cells, from 115 to 81 for WT in dually infected cells, and from 355 to 336 for DI in dually infected cells). This effect is not observed in the model, because it assumes relatively slow decay of encapsidated genomes, which may take days [52], compared to the time scale of the experiment, which takes hours. A possible explanation is inactivation of virus particles overtime, an process that has not been incorporated in our model.

Nonetheless, the cross-validation experiment showed an overall good predictive power of the model, although it underestimated the relative WT output when the DI was transfected at low DI-to-WT input ratios, i.e. 0.46, S3 Fig). In addition, full model simulations with best parameter values overestimates the number of WT encapsidated genomes in singly infected cells while the estimation is more accurate in dually infected cells (Fig 3C and 3D). This discrepancy stems from the normalization process. The WT output was normalized using single WT infection in the cross-validation analysis, resulting in underestimations of WT relative output in dual infection condition. Despite these small deviations between model prediction and experimental measurements, even with differing measurements between experiments and simulations (see Material and methods), the model is able to recapitulate WT outputs for various inputs, which reinforces its robustness and predictive ability.

## Identification of parameters affecting WT virion production

A sensitivity analysis showed that the WT/DI replication rate ($P$) and encapsidation rate ($\omega$) are the most important parameters determinind the outcome of co-infection (Fig 4A and 4B). These parameters relate to the DI design and they can be engineered to modulate WT infection. Thus, the model provides estimations on the modifications to the DI genome enhancing its replication and/or encapsidation which lead to a significant 10-fold decrease in WT virion production. This result is consistent with the hypothesis that the mechanisms of DI interference with WT replication relies on exploitation competition [1–3] for resources required for genome replication and direct competition [5] for capsid proteins.

## Impact of initial conditions

The model further predicts a crucial impact of temporal spacing and order of infection, as well as initial MOIs, on the proportion of WT virions at cell lysis (Fig 6 and S5 Fig). At equal MOIs, the DI particle needs to infect a cell within approximately a 2 hours window before or after the WT in order to be able to generate progeny, DI virions, and up to approximately 30 minutes

after the WT in order to reduce WT yields (Fig 6A). Upon co-infection, the DI particles will maximize their virion production when WT and DI initial MOIs are approximately equivalent, and the WT particles will maximize their virion production when WT MOI is larger than DI MOI (Fig 6B and 6C). At equal MOIs, the difference between DI and WT virion production is maximized at approximately simultaneous co-infection of a cell. When WT MOI is larger than DI MOI, this difference maximization is obtained when the DI infects the cell before the WT. Conversely, when DI MOI is larger than WT MOI, it is obtained when the WT infects the cell before the DI (S5 Fig).

These results are in agreement with those of [53], who found that the extent of interference, assessed by the yield of WT poliovirus, is inversely proportional to the percentage of DI in the inoculum, and that it is also affected by varying the time interval between primary and secondary infection of a cell. In their viewpoint article, [2] highlight the importance of initial conditions, such as relative initial frequencies and the time of DI emergence in a population. An earlier review [54] also emphasized this aspect, while focusing on the exclusion of one virus strain by another. Experimental studies have shown inhibition of superinfection by a resident strain, in bacteria [55] and in viruses [56, 57]. Furthermore, the DI interference efficiency may also be affected by the the ability of picornaviruses rapidly induce resistance of the host cell to superinfection [58].

Importantly, preaccumulated replicon RNAs are not trans-encapsided by capsids made from a coinfecting helper virus [59], consistent with the idea that encapsidation is coupled to RNA replication. Thus, if DI is the first to infect a cell, the genomes replicated before WT superinfection would not get trans-encapsided by WT capsids.

## Limited resource and co-evolution

The competition between WT poliovirus and DI particles within cell can be analysed in light of evolutionary game theory. For a game between WT cooperators and DI defectors, the pay-off matrix features a fitness of zero in the case of a population composed only of DIs, because they are unable to reproduce [60]. With such a feature, the evolution of a mixed WT and DI population is predicted to result in a polymorphic equilibrium, despite the greater pay-off that would result if the population was composed only of WT cooperators [60]. These strategies of cooperation and defection are common in viruses, as co-infection of the same host cell induces competition for shared intracellular products [60].

Resource availability can have important consequences on the dynamics and evolution of mixed pathogen populations [2, 61, 62]. For example, playing on resource availability could allow to slow the evolution of resistance to antimicrobial drugs [63]. In the case of DI particles, their presence within-cells infected by a WT virus is decreasing the number of resources available to the WT for replication and encapsidation, lowering WT virions production. Hence DI particles could be used to control WT infections, by lowering WT viral load, thereby facilitating further action of the immune system and/or drugs to clear the infection [46].

It has been suggested that the potential of DI particles to compete against a WT virus can be exploited to develop a new type of therapeutic antiviral strategy based on defective interfering particle competition [64]. Our model and the sensitivity analysis we have performed suggest that parameters $P$, the DI-to-WT replication ratio, and $\omega$, the DI-to-WT encapsidation ratio, are the first and second most important parameters impacting the proportion of WT virions at cell lysis. Therefore, a rational strategy to strengthen interference activity of DI genomes and thus reduce the production of WT virions is to modify DI genomes towards higher replication rate and encapsidation efficiency. Such improvements may be realized by taking advantage of the evolvability of DI genomes. Serial co-passages of WT and DI particles followed by genetic

analyses, as done in Fig 5A, 5B and 5C, would allow for the screening of other mutations providing higher replication or encapsidation of the DI. Also, the production of shorter DI genomes could lead to its faster replication.

Improving the interference at the intracellular level may cause less inhibition of WT viral load at the intercellular level, as there could be trade-offs. A reduced production of WT particles within-cells could result in a decreased MOI of WT viruses for the next infection cycle, and also to a decreased probability of a cell being co-infected. Furthermore, the narrow window of delay of co-infection for the DI to outcompete the WT as shown in Fig 6A also suggests the importance of simultaneous infection. Interestingly, recent studies show several possibilities for how co-infection is or can be favored [65–67]. Notably, the existence of vesicles containing multiple copies of virions as well as bacteria binding virions may increase the probability of simultaneous co-infection [65, 66]. Erickson et al. [67] reported that poliovirus binds lipopolysaccharide of bacteria, allowing co-infection of mammalian cells even at a low MOI. Finally, the potential of the combined DI-WT system to synergistically trigger an effective innate immune response (e.g. interferon) is also a potential avenue of investigation for the rational design of an antiviral therapy.

## Conclusion

While ecological studies for the control of pathogen populations mainly focus on preventing or slowing down the emergence of drug resistance [63, 68, 69] or on the evolution of virulence [70, 71], we take an original approach here by rather studying how to use cheater defective pathogens, competing more efficiently for shared resources, for the control of disease-inducing pathogens. Since we learned the mechanisms of intracellular interference, in a future work we would like to apply these findings for the study of the competition between WT and DI at the larger level of the tissue, embedding intracellular knowledge. It would allow us to draw guidelines to optimize DI particles at this level, based on WT viral load inhibition, and further confirm their efficiency *in vivo*.

## Materials and methods

### Competition experiment between defective genomes and wild-type genomes

**Cells.** HeLaS3 cells (ATCC CCL-2.2) provided by R. Geller and J. Frydman (Stanford University) were maintained in 50% Dulbecco's modified Eagle medium and 50% F-12 medium (DMEM/F12) supplemented with 10% newborn calf serum (NCS), 100 U/ml penicillin, 100 U/ml streptomycin and 2 mM glutamine (Invitrogen).

**Construction of viral cDNA plasmids.** The cDNA plasmid prib(+)XpA, encoding the genome of poliovirus type 1 Mahoney strain under T7-promoter and hammerhead ribozyme sequences, was reported previously [72]. Plasmid prib(+)XpA was digested by *NruI* and *SnaBI* (New England Biolabs) and ligated to produce prib(+)XpA lacking the poliovirus capsid-encoding region from 1175 to 2956 (prib(+)XpA-Δ1175–2956).

**_In vitro_ RNA transcription.** Plasmids prib(+)XpA and prib(+)XpA-Δ1175–2956 were digested by *EcoRI* or *PvuII*. Linearized plasmids were used as templates to obtain WT and DI genomic RNAs by *in vitro* transcription. *In vitro* transcribed RNAs were purified by phenol-chloroform extraction and the quality of purified RNAs was analyzed by electrophoresis on a 1% agarose gel in tris-acetate-EDTA Buffer (TAE).

**Transfection of defective interfering and wild-type genomes.** Monolayer of HeLaS3 cells was trypsinized and washed three times in D-PBS. Cells were resuspended in 1 ml D-PBS

and the number of cells were counted on a hemacytometer, followed by adjusting the concentration to $1 \times 10^7$ cells/ml. 800 $\mu$l of cells and virus RNAs (5$\mu$g of WT genomes and/or different amounts of DI genomes described later) were combined in a chilled 4-mm electroporation cuvette and incubated 20 minutes on ice. Cells were electroporated (voltage = 250 V, capacitance = 1000 $\mu$F) using Gene Pulser II (Bio-Rad), washed two times, and recovered in 14 ml prewarmed (37˚C) DMEM/F12 medium with 10% NCS. Samples were distributed on 24 well plates (250 $\mu$l/well).

Samples were collected at different time points (0, 3, 6, 9 hours for titration, and 0, 2, 3.5, 5, 7, 9 hours for RNA extraction) after electroporation. For titration and evaluation of encapsidated RNAs, samples were then frozen and thawed three times, followed by centrifugation at 2,500 g for 5 minutes, and supernatants were collected. Samples for evaluation of encapsidated RNAs were further treated with mixture of RNase A (20 $\mu$g/ml) and RNase T1 (50 U/ml) (Thermo Fisher Scientific) for three hours. Samples were stored at −80˚C.

**Titration of virus samples.** Monolayers of HeLaS3 cells in 6-well plates were infected with 250 $\mu$l of serially diluted virus samples at 37˚C for 1 hour and then overlaid with DMEM/F12 including 1% agarose. After 48 hours of infection, infected cells were fixed by 2% formaldehyde and stained by crystal violet solution. Titers were calculated by counting the number of plaques and multiplying their dilution rates.

**RNA extraction.** 250 $\mu$l of samples was added to 750 $\mu$l of TRI-reagent LS (Sigma Aldrich), and RNAs were extracted following the kit protocol. Briefly, 200 $\mu$l of chloroform was added to each sample, shaken vigorously, and incubated at room temperature for 10 minutes. Then samples were centrifuged at 12,000 g for 15 minutes at 4˚C. The upper aqueous phase was transferred to a fresh tube and 0.5 ml of isopropanol was added. After incubation at room temperature for 10 minutes, samples were centrifuged at 12,000 g for 8 minutes at 4˚C to precipitate RNAs at the bottom of the tube. The supernatant was removed and the residue was washed by 1 ml of 75% ethanol. After centrifugation at 7,500 g for 5 minutes at 4˚C, the RNA pellets were dried for 5–10 minutes. RNAs were resuspended in nuclease-free water.

**Reverse transcription.** 2.5 $\mu$l of RNA samples was mixed with 0.5 $\mu$l of 2 $\mu$M primer (5'-CTGGTCCTTCAGTGGTACTTTG-3'), 0.5 $\mu$l of 10 mM dNTP mix, and 2.5 $\mu$l of nuclease-free water. Samples were incubated at 65˚C for 5 minutes, and then placed on ice for 1 minute. After adding 10 $\mu$l of cDNA synthesis mix (1 $\mu$l of 10× RT buffer, 2 $\mu$l of 25 mM MgCl2, 1 $\mu$l of 0.1 M DTT, 1 $\mu$l of RNaseOUT and 1 $\mu$l of Superscript III RT enzyme), samples were incubated at 50˚C for 50 minutes, and then at 85˚C for 5 minutes to terminate reactions. 1 U of RNase H was added to each sample, followed by incubation for 20 minutes at 37˚C. Then, 0.1 U of Exonuclease I was added to each sample, followed by incubation at 37˚C for 30 minutes and at 80˚C for 15 minutes to terminate reactions. cDNA samples were stored at −20˚C.

**Design of primers and Taqman probes.** Primers and Taqman probes for droplet digital PCR assay were designed with PrimerQuest Tool (Integrated DNA Technologies). The primers and probe for WT genomes are 5'-CCACATACAGACGATCCCATAC-3', 5'-CTGCCCAGTGTGTGTGTAGTAAT-3', and 5'-6-FAM-TCTGCCTGTCACTCTCTCCAGCTT-3'-BHQ1. The primers and probe for DI genomes are 5'-GACAGCGAAGCCAATCCA-3', 5'-CCATGTGTAGTCGTCCCATTT-3', and 5'-HEX-ACGAAAGAG/ZEN/TCGGTACCACCAGGC-3'-IABkFQ.

**Droplet digital PCR assay.** 2 $\mu$l of serially diluted cDNA samples was mixed with 10 $\mu$l of 2× ddPCR supermix for probes, 1 $\mu$l of 20× WT primers/probe, 1 $\mu$l of 20× DI primers/probe, and 6 $\mu$l of nuclease-free water. 20 $\mu$l reaction mix of each sample was dispensed into the droplet generator cartridge, followed by droplet production with QX100 droplet generator. Then PCR was performed on a thermal cycler using the following parameters: 1 cycle of 10 minutes at 95˚C, 30 cycles of 30 sec at 94˚C and 1 minute at 60˚C, 1 cycle of 10 minutes

at 98˚C, and held at 12˚C. Positive and negative droplets were detected by QX100 droplet reader.

## Model reduction

As our mathematical model (Eqs 1–4) presents a classical problem of parameter identifiability, we built a lower dimensional model to solve this problem by assuming that the decrease in resources due to viral uptake for replication follows a logistic decreasing function. This assumption was verified by analyzing the curves of $R(t)\theta\varepsilon$ as a function of time on a first set of "blind" optimizations. Thus, we can recast the model using the following lower dimensional description:

$$\frac{dG_{WT}}{dt} = \Lambda(t)G_{WT} - c_g\kappa CG_{WT} - \alpha G_{WT} \tag{7}$$

$$\frac{dC}{dt} = \eta\Lambda(t)G_{WT} - \kappa(G_{WT} + \omega G_{DI})C - \beta C \tag{8}$$

$$\frac{dG_{DI}}{dt} = P\Lambda(t)G_{DI} - \omega c_g\kappa CG_{DI} - \alpha G_{DI} \tag{9}$$

$$\Lambda(t) = \frac{L}{1 + e^{s(t-t_0)}} \tag{10}$$

The logistic function (Eq 10) is characterized by the curve's maximum value $L$ and steepness $s$, and the time of the sigmoid's midpoint $t_0$. While this reduced version only decreases the number of parameters to be estimated by one ($L$, $s$ and $t_0$ instead of $\theta$, $\varepsilon$, $\lambda$ and $\gamma$), it partially solves the identifiability problem by removing biologically interpretable parameters and just assuming a logistic function for resource uptake and replication. An analytical justification of this model choice based on time scale separation can be found in the S1 Text.

## Fit to experimental data

The model was fitted to the experimental data of Fig 1C and 1D in order to estimate model parameters describing our biological system. De novo RNA replication can only be detected around 2 hours post transfection [31, 73–75]. We thus used experimental data from 2 hours post transfection for parameter estimation. Indeed, there are several steps of poliovirus infection cycle before replication can start, including translation of positive-sense genomes [48] and transition from a linear, translating RNA to a circular RNA competent for replication [31, 73–75]. As our model does not account for those first steps, we only used experimental data from 2 hours post transfection for parameter estimation.

Raw experimental data are WT and DI total RNA copy number ($g_{WT}^{tot}$ and $g_{DI}^{tot}$, Fig 1C) and RNAse treated genome copy number ($v_{WT}$ and $v_{DI}$, Fig 1D). The former corresponds to the total number of genomes (naked and encapsidated) and the latter to the number of encapsidated genomes. The numbers of WT and DI naked (i.e. non-encapsidated) genomes are thus: $g_{WT} = g_{WT}^{tot} - v_{WT}$ and $g_{DI} = g_{DI}^{tot} - v_{DI}$. Additionally, as raw data was obtained at the cell population level, it was normalized by the average number of successfully transfected cells in order to get the average number of naked and encapsidated WT and DI genomes per cell.

In all, experimental data comprises three replicates of independent populations sampled at 2, 3.5, 5, 7 or 9 hours post transfection. Three different conditions were tested: (i) cells

dually transfected by WT and DI genomes, (ii) cells transfected by WT genomes only and (iii) cells transfected by DI genomes only. Transfected volumes of WT and DI genomes were calibrated to a ratio of WT:DI = 4:1 in order to approximately obtain a ratio of 1:1 after transfection.

Parameter estimation was achieved through nlminb optimization function in R software [76] embedded in an iterative process. Each optimization consisted in minimizing the sum of the squared errors between experimental and simulated normalized data points for all variables and conditions. The least square function is as:

$$
\begin{aligned}
LS \quad = \quad & \sum_r \sum_t \underbrace{\left[ (\bar{G}_{WT}(t) - \bar{g}_{WT}(t,r))^2 + (\bar{G}_{DI}(t) - \bar{g}_{DI}(t,r))^2 \right]}_{\text{Dually transfected cell}} \\
& + \sum_r \sum_t \underbrace{\left[ (\bar{C}_{WT}(t) - \bar{v}_{WT}(t,r))^2 + (\bar{C}_{DI}(t) - \bar{v}_{DI}(t,r))^2 \right]}_{\text{Dually transfected cell}} \\
& + \sum_r \sum_t \underbrace{\left[ (\bar{G}_{WT}(t) - \bar{g}_{WT}(t,r))^2 + (\bar{C}_{WT}(t) - \bar{v}_{WT}(t,r))^2 \right]}_{\text{WT genome transfected cell}} + \sum_r \sum_t \underbrace{\left[ (\bar{G}_{DI}(t,r) - \bar{g}_{DI}(t,r))^2 \right]}_{\text{DI genome transfected cell}}
\end{aligned}
\tag{11}
$$

with $r$ indicating the replicate number and $t$ the sampling time ($t$ = 3.5 to 9 hours post transfection). Initial conditions for all variables in all infection conditions were obtained from averaging experimental observations over the 3 replicates at $t_0$ = 2 hours post transfection. In Eq (11), we define:

$$
\bar{x}_i = \frac{\log(x_i + 1)}{\max(g_i)} \; , \; \bar{y}_i = \frac{\log(y_i + 1)}{\max(v_i)}
$$

with $x$, resp. $y$, being either experimental ($g$, resp. $v$) or numerical ($G$, resp. $C$) naked, resp. encapsidated, genomes data and $i$ for WT or DI.

The iterative process was applied as follows (see also S4 Fig for a schematic representation). Boundaries on parameter values were defined based on a first set of "blind" optimizations, the intervals still remaining large and realistic. For each parameter $p$, let us denote $p_{min}$ the lower boundary and $p_{max}$ the upper boundary. For the first iteration, random values of parameters were drawn from uniform distributions, as $p_{start} \sim \text{Unif}(p_{min}, p_{max})$, defining the starting point for optimization. Let us denote $\tilde{p}$ the optimized parameter value. In subsequent iterations, the starting point for each parameter was then randomly drawn from a uniform distribution, as $p_{start} \sim \text{Unif}(\max(0.95 \cdot \tilde{p}, p_{min}), \min(1.05 \cdot \tilde{p}, p_{max}))$. In all, 20 iterations were conducted, and this iterative procedure was implemented 250 times, each time with a different random starting point in the first iteration. Thus, $250 \times 20 = 5000$ optimizations were performed in total.

The goodness of fit was evaluated by ordinary least square (Eq (11)) and sum of residuals $R^2$ between experimental and simulated normalized data.

This optimization procedure was applied in two steps. In the first step, the optimization was performed on the reduced version of the model (Eqs 5–10), thus estimating nine parameters: the six parameters corresponding to (i) the DI ($P$ and $\omega$), (ii) the production of capsids ($\eta$), (iii) the encapsidation process ($\kappa$) and (iv) the decay rates of genomes and capsids ($\alpha$ and $\beta$); and the three parameters of the logistic function representative of the time-dependent replication rate ($L$, $s$ and $t_0$). In the second step, the six redundant parameters between the reduced and full version of the model ($P$, $\omega$, $\eta$, $\kappa$, $\alpha$ and $\beta$) were fixed to their best estimated value obtained during the first step. The remaining four parameters ($\theta$, $\epsilon$, $\lambda$ and $\gamma$) were estimated by optimizing the full version of the model (Eqs 1–6). The first 85 best sets of parameters that

provide the most accurate fits to the experimental data are used for sensitivity analysis of the full model (Table 1 and Figs 4, 5 and 6, S3 and S5 Figs)) and to confirm the theoretical validity of the reduced model (Fig 3 and S2 Fig, and S1 Table).

**DI variant identification, CirSeq and Analysis of allele frequencies.** Monolayer of HeLaS3 cells was trypsinized and washed three times in D-PBS. Cells were resuspended in 1 ml D-PBS and the number of cells were counted on a hemacytometer, followed by adjusting the concentration to $1 \times 10^7$ cells/ml. 800 $\mu$l of cells and virus RNAs (5$\mu$g of poliovirus type 3 WT genomes and 5$\mu$g of DI genomes) were combined in a chilled 4-mm electroporation cuvette and incubated 20 minutes on ice. Cells were electroporated as described above. Supernatants containing PV3 WT virus and trans-encapsidated DI genomes were collected 24 hr post transfection and used to infect a fresh plate of HeLa cells. The infection procedure was repeated for 8 passages.

For preparing CirSeq libraries, each passaged virus ($6 \times 10^6$ PFU) was further expanded in parental cells seeded in four 150 mm dishes. The culture medium was harvested before the appearance of severe CPE, and the cell debris was removed by centrifugation at 3,000 rpm for 5 min. The virion in the supernatant was spun down by ultracentrifugation at 27,000 r.p.m, 2 hours, 4°C and viral RNA was extracted by using Trizol reagent. Each 1$\mu$g RNA was subjected to CirSeq libraries preparation as described previously [44, 45]. The CirSeq pipeline allows error detection in RNAseq through consensus generation and quality filtering to overcome the intrinsic error rate associated with reverse transcription. The experimental and computational protocols are described in detail previously [44, 45]. Briefly, purified polyA+ RNA from infected cells is fragmented to yield 80–100 bp fragments, circularized, and subject to rolling-circle reverse transcription. This procedure yields tandem reverse transcripts that are used to correct reverse transcription errors. Variant base-calls and allele frequencies were then determined using the CirSeq v2 package (https://andino.ucsf.edu/CirSeq). Circularized repeats are oriented to the reference genome and variants are called from raw reads based on phred33 scores of 20. These tandem variant-called reads are then aligned to each other to generate consensus sequences with a theoretical error of $10^{-6}$. Technical replicates of passaged libraries, and individual sequencing lanes, were compared after CirSeq mapping and pooled for analysis of fitness. Raw reads are deposited at Bioproject PRJNA669406. All consensus, and mapped reads from CirSeq are available at https://purl.stanford.edu/gv159td5450. Positive selected mutations within the DI genome, which frequency increased over passages, were selected and analyzed for replication fitness and competition with WT virus (Fig 5D and 5E).

**Variants passaging experiment.** A PV1-DI construct encoding the Venus (a green fluorescent protein) gene in place of the P1 gene was used for the following experiments. We transfected 5$\mu$g of PV3 and 1.25$\mu$g of PV1-DI genomes, and collected viruses at 24 hours after transfection as the passage 1 (P1) sample. Then, $1.0 \times 10^6$ HeLaS3 cells were infected with the different amounts of the P1 sample (500, 100, and 5 $\mu$l), and collected 24 hours after infection. The virus samples were passaged 8 times in the same manner. Viral RNAs for each sample were then analyzed by CirSeq [44] to identify accumulated mutations in DI.

**Competition experiment and analysis.** An additional experiment was conducted on eight DI mutants and the parental PV1-DI construct encoding the Venus gene. To obtain DI particles for the experiments, a packaging cell line was established. HeLaS3 cells were transfected with a pcDNA4 plasmid encoding the PV1 P1 (capsid) gene, followed by selection with Zeocin. We used a clone stably expressing P1 proteins (HeLaS3/P1) to generate DI particles. HeLaS3/P1 cells were transfected with DI RNAs by electroporation, and DI particles were collected 24 hours after transfection. Each of the DI variant was put in competition with the WT virus, and the number of naked genome copies was measured at 3, 4.33, 6.33 and 8.17 hours

post transfection, with three replicates per time-point. Only the ratio of DI-to-WT naked genome copies is kept for further analysis.

We assumed that the mutations only affect the replication of each DI variant, hence we set all model parameters to their best estimated values (see Table 1) except for the DI-to-WT replication ratio $P$ that we varied between 1 and 1.8. For simplicity, we assumed that the ratio of DI to WT encapsidation rate ($\omega$) remained the same as estimated from the main experiment, as we showed that $P$ is a more sensitive parameter than $\omega$ (Fig 4). The full model was used for this analysis (Eqs 1–6). We set the initial conditions to 10 copies of DI and WT (each) naked genomes at 2 hours post transfection. We assumed that there was initially no capsids or encapsidated genomes, and the number of resource units was set as in the main experiment to $\lambda/\gamma$. We recorded the simulated numbers of DI and WT naked genome copies at the experimental time-points for varying values of $P$ (step of 0.01). The best $P$ value for each DI variant was found as the one minimizing the sum of squared differences between experimental and simulated ratio of DI-to-WT naked genome copies.

## Model predictions

**Cross-validation.** We cross validate the results of our optimization procedure by assessing how well the model is able to predict the relative WT virus burst size for various WT to DI initial ratios (after transfection) for which it has not been trained. We first obtain an optimal set of model parameters on our time series experimental data ($g$ and $v$) featuring initial WT: DI = 1:1. We then test five additional DI-to-WT initial ratios, ranging from 0 to 3.6. Initial conditions for model simulations were set as the average of experimental values for each of the five initial ratios.

In the cross-validation experiment, evaluation of the relative WT virus burst size was based on the count of plaque-forming units (PFUs). In the time-series experiment that was used for parameter estimation, the number of WT virions ($v_{WT}$) was estimated by digital droplet PCR. Assuming that the ratio of WT infectious to non-infectious particles and the multiplicity of infection (MOI) of WT virus are both constant independently of initial conditions, the relative PFU of WT virus for each initial condition should be a good proxy of the relative WT burst size.

The experiment was conducted on a cell population, and then the measurements were normalized by the number of successfully transfected cells. In some cases, the average experimental MOIs were small, potentially leading to not all cells being co-infected by WT and DI genomes. We integrated this aspect in our simulated burst size calculations, with the probabilities that a cell would be infected by both DI and WT genomes or only by WT genomes. We assume that the number of DI genomes infecting a cell $X_{DI}$ results from a Poisson distribution of parameter the average DI MOI $n_{DI}$, as $X_{DI} \sim Pois(n_{DI})$. The probability that no DI genome enters a cell is thus $P(X_{DI} = 0) = e^{-n_{DI}}$. Conversely, the probability that at least one DI genome enters a cell is $1 - e^{-n_{DI}}$. The expressions are equivalent for the WT virus. Let us denote the WT burst size in WT-DI infection as $\mathcal{B}_{WT}(WT - DI)$ and the WT burst size in WT-only infection as $\mathcal{B}_{WT}(WT)$. We weighted WT simulated burst sizes as follows: $\mathcal{B}_{WT} = (1 - e^{-n_{DI}})(1 - e^{-n_{WT}})\mathcal{B}_{WT}(WT - DI) + e^{-n_{DI}}(1 - e^{-n_{WT}})\mathcal{B}_{WT}(WT)$.

WT PFU experimental values and WT burst size model predictions ($\mathcal{B}_{WT}$) were normalized for all initial ratios by their respective values in the absence of DI genome (i.e. DI-to-WT initial ratio of 0). For the experimental data the average over the three replicates was taken for normalization. The performance of the model to predict relative WT burst size was evaluated by $R^2$ and p-value of a Pearson correlation test between experimental and simulated datapoints.

**Sensitivity analysis.** A sensitivity analysis was performed to assess the relative importance of each parameter on the proportion of WT virions at cell lysis, defined as:

$$\Phi_{WT} = \frac{\mathcal{B}_{WT}}{\mathcal{B}_{WT} + \mathcal{B}_{DI}} \tag{12}$$

Based on parameter estimation, each parameter was approximately varied by ±50% of its best estimated value. Based on these boundaries, each parameter was allocated a vector of five equidistant values (except for $\alpha$ that was not varied because it was estimated at 0, see Table 1). Then, all distinct combinations of parameter values were tested according to a full factorial design. In all, $5^9 = 1,953,125$ simulations of the full model were performed. All simulations started at time 0 hours post transfection with 10 copies of WT and DI genomes ($G_{WT}(0) = G_{DI}(0) = 10$), no capsids nor encapsidated genomes ($C(0) = C_{WT}(0) = C_{DI}(0) = 0$), and $R(0) = \lambda/\gamma$. Simulations were conducted until 9 hours post transfection. An analysis of variance (ANOVA function in R software) was then conducted to assess the importance of each parameter and their second-order interactions on the variance of $\Phi_{WT}$.

**Impact of delay and multiplicities of infection on WT and DI burst sizes.** In the experiment, WT and DI genomes were co-transfected to cells and in quantities yielding an MOI ratio of approximately WT:DI = 1:1. We conducted two sets of simulations to study the impact of varying either (i) the time between cell infection by WT and DI or (ii) the MOIs of WT and DI on their burst sizes. All the simulations were conducted on the full version of the model (Eqs 1–6). In the first set of simulations, WT and DI burst sizes were recorded for various delays between primary and secondary infection of a cell, ranging from -7 to +7 hours for the time of DI infection compared to the WT. The MOI upon infection of the cell was set to 10 for both WT and DI (i.e. $G_{WT}(0) = 10$ genomes and $G_{DI}(t_d) = 10$ genomes, with $t_d$ the delay for DI infection), the number of capsids and encapsidated genomes to 0 and $R(0)$ to $\lambda/\gamma$. In the second set of simulations, WT and DI burst sizes were recorded for various WT and DI initial MOIs, ranging from 0 to 1000 virions. All the other variables were set as described for the study of delays. Then, WT burst sizes were normalized by the WT burst size corresponding to WT:DI = 1:0 initial MOIs (i.e. infection by the WT virus only at low MOI), and DI burst sizes by the DI burst size corresponding to WT:DI = 1:1 initial MOIs (i.e. infection by both WT and DIs at low MOI).

## Supporting information

**S1 Fig. Experimental data and $\mathcal{M}^{12}$ model fit for the naked and encapsidated wild-type (WT) and defective interfering (DI) genomes, with same encapsidation rate.** Evolution of the number of WT and DI (A-B) naked genome copies and (C-D) encapsidated genome copies with time, from 2 to 9 hours post transfection (hpt). (A & C) show data in dually transfected cells whereas (B & D) show data in singly transfected cells. WT and DI results are shown in blue and red color, respectively. Crosses indicate experimental data for 3 replicates per sampling time at 2, 3.5, 5, 7 and 9 hpt. Solid curves show the fit of the full model $\mathcal{M}^{12}$ with same encapsidation rate for WT and DI naked genomes (Eqs. (S10)–(S13) in S1 Text).
(PDF)

**S2 Fig. Histograms and correlations of the genome replication factor ($\theta$), the resource linear production ($\lambda$), the decay rate ($\gamma$), and the resource capture rate ($\varepsilon$).** The best 123 estimated values for each parameter are represented. A-D: Histograms of best estimated values for $\theta$ (A), $\varepsilon$ (B), $\lambda$ (C) and $\gamma$ (D). E: Correlation between resource production to decay rate and genome replication factor. The best fit curve is shown in red and its equation is provided

(Pearson p-value $<2.2 \cdot 10^{-16}$ and $R^2 = 0.897$). F: Time-dependent replication rate given by the reduced model ($\Lambda$ (t), dashed line) and the full model ($\theta \varepsilon R(t)$, plain line).
(PDF)

**S3 Fig. Cross-validation of the model (Eqs (1)–(6)).** Three experimental replicate values (black dots) of relative WT virus output are represented for various DI to WT input (proxy of multiplicities of infection) ratios. Red dots indicate predicted relative WT output starting with the same experimental input ratios. Experimental WT output corresponds to PFU while simulated WT output corresponds to burst size (number of encapsidated genomes at 9 hours post infection). All outputs were normalized by the output value (or the mean for experimental data) of WT:DI = 1:0 input ratio. R-squared and p-value of a Pearson correlation test between experimental and predicted WT outputs are given in the graphic.
(PDF)

**S4 Fig. Diagram of the parameter fitting procedure.**
(PDF)

**S5 Fig. Model-predicted delay at various multiplicities of infection (MOIs).** A: Predicted impact of delay for DI particle infection of a cell (x-axis, in hours) on WT (blue) and DI (red) burst sizes (y-axis). One line represents one DI MOI and one column one WT MOI. The grey vertical line indicates no-delay (simultaneous infection). The red vertical line indicates the peak of DI burst size and the green vertical line the maximum difference of DI to WT burst size. B: Heat map of predicted delay for the maximum difference of DI to WT burst size (green lines in A). C: Heat map of predicted delay for WT and DI burst size curves intersection.
(PDF)

**S6 Fig. Experimental data and reduced model fit for the naked and encapsidated wild-type (WT) and defective interfering (DI) genomes, with *P* fixed to the ratio of WT to DI genome lengths.** Evolution of the number of WT and DI (A-B) naked genome copies and (C-D) encapsidated genome copies with time, from 2 to 9 hours post transfection (hpt). (A & C) show data in dually transfected cells whereas (B & D) show data in singly transfected cells. WT and DI results are shown in blue and red color, respectively. Crosses indicate experimental data for 3 replicates per sampling time at 2, 3.5, 5, 7 and 9 hpt. Solid curves show the fit of the reduced model with *P* fixed to the ratio of WT to DI genome lengths (*P* = 7515bp/5733bp, Eqs (7)–(10)).
(PDF)

**S1 Table. Model selection.** For each model, described in S1 Text, the goodness of fit is evaluated by the squared Pearson correlation coefficient between experimental and fitted data ($R^2$), and the quality of the models is evaluated by the log-likelihood ($-2 \cdot \log(L)$) and Akaike information criterion (AIC) from a linear model between experimental and fitted data.
(ZIP)

**S1 Text. Model selection procedure and analytical study of the model.**
(PDF)

## Acknowledgments

We thankfully acknowledge Judith Frydman and the members of the Andino and Bianco lab for helpful discussions.

## Author Contributions

**Conceptualization:** Yuta Shirogane, Elsa Rousseau, Jakub Voznica, Igor M. Rouzine, Simone Bianco, Raul Andino.

**Data curation:** Yuta Shirogane, Elsa Rousseau, Jakub Voznica.

**Formal analysis:** Elsa Rousseau, Jakub Voznica, Igor M. Rouzine, Raul Andino.

**Funding acquisition:** Simone Bianco, Raul Andino.

**Investigation:** Yuta Shirogane, Yinghong Xiao, Weiheng Su, Adam Catching, Zachary J. Whitfield.

**Methodology:** Yuta Shirogane, Elsa Rousseau, Jakub Voznica, Yinghong Xiao, Weiheng Su, Adam Catching, Zachary J. Whitfield, Igor M. Rouzine, Simone Bianco, Raul Andino.

**Project administration:** Simone Bianco, Raul Andino.

**Resources:** Simone Bianco, Raul Andino.

**Software:** Elsa Rousseau, Jakub Voznica.

**Supervision:** Igor M. Rouzine, Simone Bianco, Raul Andino.

**Validation:** Yuta Shirogane, Elsa Rousseau, Jakub Voznica, Yinghong Xiao, Weiheng Su, Adam Catching, Zachary J. Whitfield, Igor M. Rouzine.

**Visualization:** Yuta Shirogane, Elsa Rousseau, Raul Andino.

**Writing – original draft:** Yuta Shirogane, Elsa Rousseau, Simone Bianco, Raul Andino.

**Writing – review & editing:** Yuta Shirogane, Elsa Rousseau, Jakub Voznica, Yinghong Xiao, Weiheng Su, Adam Catching, Zachary J. Whitfield, Igor M. Rouzine, Simone Bianco, Raul Andino.

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
