## [Decision Letter · Decision Letter 0]

20 Feb 2021

Dear Raul,

Thanks very much for submitting your manuscript "Experimental and mathematical insights on the interactions between poliovirus and a defective interfering genome" for consideration at PLoS Pathogens. As with all papers reviewed by the journal, your manuscript was reviewed by members of the editorial board and by three independent reviewers. In light of the reviews (below this email), we would like to invite the re-submission of a significantly-revised version that takes into account the reviewers' comments.

We cannot make any decision about publication until we have seen the revised manuscript and your response to the reviewers' comments. Your revised manuscript is also likely to be sent to reviewers for further evaluation.

Please prepare and submit your revised manuscript within 60 days. If you anticipate any delay, please let us know the expected re-submission date by replying to this email. Please note that revised manuscripts received after the 60-day due date may require evaluation and peer review similar to newly submitted manuscripts.

Best regards,

Bert Semler

Guest Editor

PLOS Pathogens

Adolfo García-Sastre

Section Editor

PLOS Pathogens

Kasturi Haldar

Editor-in-Chief

PLOS Pathogens

orcid.org/0000-0001-5065-158X

Michael Malim

Editor-in-Chief

PLOS Pathogens

orcid.org/0000-0002-7699-2064

Reviewer's Responses to Questions

**Part I - Summary**

Reviewer #1: There is a renewed interest in defective interfering (DI) particles (and other types of defective genomes) because they are produced and maintained in populations of many viruses (particularly RNA viruses), and because they may play regulatory roles, particularly in the interaction with the host immune system. In the present study, experimental data on WT and DI poliovirus are presented, and confronted with a relatively simple model (a reduced model to determine some parameters and a full one) of differential equations, based on one previously developed for HIV-1. The model considers as central parameters rates of replication and encapsidation to explain the outcome of WT-DI competitions. The predictive value of the model is tested with some experimental results, and reasons for agreements and disagreements are discussed.

Although the study is of potential interest, in its present form it fails to relate the findings to previous work on DI particles, and models on interference mechanisms. The authors fail also to underline the major conclusions, novelty, and implications of the experimental results and model. In addition, some experimental protocols should be described in greater detail.

1. Other previously described models and their predictions should be compared with the differential equation-based model presented in the study (Frank, J. Theor. Biol. 2000; Stauffer Thompson et al. J. Gen. Virol. 2009; Laske et al. Virus Res. 2016; Kirkwood and Bangham, PNAS, 1994; among others). For example, the deterministic model of Kirkwood and Bangham emphasizes unpredictable effects of DI activity, a point not discussed in the present study. The merits of the new model relative to previous ones, and its advantages for future work should be explained.

2. According to Figure 5 legend the poliovirus used was type 3 (PV3), while according to the Supplementary Methods the WT was PV3 while the DI was constructed on PV1. The poliovirus used as WT and for DI construction should be explained and, if different viral types were used, a justification is needed. Also, how the present DI structure compares with others previously characterized for poliovirus should be indicated.

3. A critical experimental value is the proportion of encapsidated RNA measured by resistance to a RNase mixture. Controls (or literature references of previous work by the authors where the controls are described) consisting of mixtures of encapsidated (protected) purified particles and free viral RNA (unprotected) to evaluate the reliability of the quantifications should be described. Indicate also the proportion of cells that were transfected.

4. There is some confusion about what the authors mean by “replication rate”. Having a shorter RNA, DIs are expected to complete a round of template copying earlier than WT RNA. This does not imply that the replicase (or replicative complex or organelle) travels faster along the RNA template when copying DI RNA than WT RNA. Unless a different rate of polymerase progression has been documented, it would be better to refer to replication being completed in a shorter time for DI than WT. An additional ambiguity becomes apparent when the authors state that “replication rate” in their model includes also translation of the positive sense RNA genome. To complicate matters, in section 4.1 the authors suggest that the DIs will replicate faster than the WT by a factor given by the ratio of WT to DI genome lengths, but the observed discrepancy may be attributed to the various processes linked to replication (which processes in addition to translation mentioned earlier?). In this section they also refer to “replication speed”.

5. The data presented in Fig. 5 require additional information. In panel A, the passages produced a number of mutant DIs. The reader assumes that no mutations occurred in WT but this should be stated. Fitness of variant DIs is mentioned but no information on their values and how was fitness defined and determined is provided. DI mutant 1952 in panel D is named 19522 in panel C. The meaning of the volumes given in panel B PV3 should de explained. The differences between panels E and F and their significance are not clear.

6. While the Results and figure legends will benefit from expansion, the Discussion is very long and with speculative arguments. At times, it is considerably naïve, for example when referring to evidence of dengue coinfections in India, when the literature includes many examples of coinfections. It is unlikely that the non-specialist reader can capture the main conclusions and novelty of the present study among a quite disorganized amount of information, mainly in the Discussion.

7. At times the distinction between what is predicted by the model without experimental verification and what has been experimentally studied is not clear. While the data in Figure S2 are an experimental verification of model predictions, those in Figures 6 and S4 do not seem to have an experimental verification with the poliovirus system. If this is correct, it should be indicated.

Reviewer #2: Summary:

The presented manuscript `Experimental and mathematical insights on the interactions between poliovirus and a defective interfering genome’ aims to investigate the intracellular time course dynamics of poliovirus and its defective viral genomes (DVGs) that interfere with the full-length virus (FLV) via two mechanisms. First, DVGs are replicons which need FLV’s capsid to form particles, thus hampering encapsidation of the FLV. Second, DVGs replicate more rapidly, thus depleting cellular resources at the expanse of the FLV. Briefly, the authors used a mathematical model to fit the time course genome concertation data of FLV and DVGs to estimate the model parameters. They conclude that the most important parameters that determine the outcome of FLV and DVG competition are DVG-to-FLV replication ratio, P, and DVG-to-FLV encapsidation ratio, omega. The authors then use additional experimental data to validate their model predictions.

Strengths/Weaknesses:

I personally find the idea of combining the experimental data on FLV and DVGs with mathematical modelling interesting and sufficiently novel. The study addresses a rather unexplored problem of numerical quantification of DVG parameters during the FLV-DVG competition as these could help to better design treatment protocols should DVG research progress to a phase of therapy development in the future, particularly the optimal dose and the time of DVG administration. On the other hand, successful DVG infection would have to be secured. Cell transfection is a limitation and it is unclear to me whether FLV-DVG multiple-cycle infection dynamics would be possible to track by the authors. The authors also present their assumption about the model based on the knowledge of their DVGs as a result claiming that the model predicts P and omega to be greater than 1 (e.g. Discussion, lines 386-387). If P=omega=1, there’s no competitive advantage and the outcome will depend on the initial conditions/time of delivery. Computational methods need to be described in more detail.

Reviewer #3: PPATHOGENS-D-21-00004_reviewer

Summary

(Given how covid has changed my working life, I'm reviewing this as I would review a preprint I need. As in, does it make sense, is it basically sound, and do I understand what they're saying? If yes to all three, then I don't think it needs much revision. So I read carefully, but I'm not looking to nitpick in this review.)

This is a really nice example of model and experiment working together. The question of interfering particles is important to understanding viral fitness, the experimental methods are clean, and the modeling is insightful and simple but not-too simple and directly relevant to the experiments. And the model makes successful, non-trivial out-of-sample predictions. Any modeling paper that actually predicts something it didn't fit to is good work (and much more rare than it should be!)

I'm familiar with the both the virology and model, but much better equipped to critique model methods than lab. The model is reasonable, as is the optimization procedure to fit it, and the results are clearly communicated.

In general, the paper is clearly written. I felt like I knew exactly what the paper was about from the intro and why you were interested in it on the first read (with one minor caveat below). This impression held up through the rest of it.

**Part II – Major Issues: Key Experiments Required for Acceptance**

Reviewer #1: See Part I

Reviewer #2: The presented mathematical model is a modification of a model from Rouzine and Weinberger (2013) in which a new variable was introduced to simulate cellular resource depletion by FLV and DVGs. I have the following questions regarding modelling, and I list them point by point:

1) Have the authors tried to fit the model with a fixed P? There is a note in the manuscript that the estimated ratio could correspond to 1.311 based on the FLV and DVG genome lengths. This would lower the dimensionality of the fitting problem. How would the predictions change?

2) What is the reason to assume that DVGs encapsidate faster than FLV? How do the predictions change if no such competitive advantage is assumed for DVGs?

3) When fitting the model, both naked and encapsidated genome abundancies define the objective function to be minimized by least squares. I am not an experimentalist and don’t understand the protocol much, but how do the authors differentiate between encapsidated genomes and empty capsids (e.g. Figure 3)? Are the data in Fig. 3 truly just encapsidated genomes? I think this is important to also write in the text as modellers could struggle to find this information in there.

4) What is the authors’ opinion on the possible assumption that DVG uses less cellular resources per genome than FLV, possibly due to its smaller size? Is that something they would consider in their model?

5) Line 222: Predictions also demonstrate that DVGs are more efficiently encapsidated than FLV (omega=2.185). I disagree with this sentence because omega>1 is the assumption the authors make, not a prediction. Can the model reproduce the data when omega=1? See my comment above.

6) Line 226-227: Again, not a prediction, interference is an assumption in the model which allows to fit the data.

7) In the equation (10) for Lambda(t), the factor s should have units of min^-1 and not be unitless as (t-t0) is in mins and L is unitless. Or is it L that should have units?

8) To be honest, I don’t follow the model reduction procedure. When I look at the models (1)-(4) and (7)-(10), they should at least yield the same units, that is genomes/time or capsid/time. When I look at eq. (2) for capsid dynamics, then e.g. the rate (eta x Lambda(t) x G_wt) has units (capsid x genome^-1 x nothing x genome) which results in just capsid as units, given Table 1. For this reason, the rates are not comparable. Also, lines 236-237, these time-dependent rates 3.07x10^-2 and 3.02x10^-2 are unitless?

9) Figure S2: Is this a reasonable validation? This is related to my question about free capsids – if the quantification of encapsidated genomes accounts for free capsids, PFUs might not correspond and normalization will mask this.

10) How exactly are the data from Figure 1 used in modelling? This part is a bit unclear to me.

11) Figure 5: What exactly is this figure saying? If I understand correctly, 8 DVGs were selected and transfected with the FLV and the DVG/FLV ratio was determined over the time. The DVG/FLV ratio at 8h post transfection was plotted against the model (which one?) prediction of P for each variant to demonstrate that higher DVG/FLV ratio implies higher P. Can you say the same for omega?

12) What were the criteria for those 8 DVGs to be selected? Have the authors used any bioinformatics methods or tools for variant calling? If so, they should be detailed in the Methods section, too.

13) Since I am not an experimentalist, it is difficult for me to assess which experiments can and cannot be technically done. Figure 6A and S4 show the impact of different time of delivery of DVGs on the FLV growth. If repeated transfection is possible, experiments to validate these predictions are desirable.

14) Lines 460-461: The authors write “This decay might also be real, but we chose not to include it in our model because of the too large number of parameters it has.” It is not acceptable to kill a parameter because it’s inconvenient.

Reviewer #3: Major comments

The idea to use the reduced model to pin down an unidentifiability before going back to the full model is clever, but I suspect the authors could go further with this line of reasoning. I interpret the fact that the reduced model works well (in fact slightly better than the full model) says G_wt(t) + G_di(t) is to an excellent approximation a linear function of R(t), such that dR/dt reduces to the logistic differential equation. Which is to say that the sum of Gs is very fast on the timescale R evolves, or that d(G_wt+G_di)/dt >> dR/dt . I don't quite have it in me tonight to formally work through this slow/fast separation of scales argument, but the authors might do so to better understand why the reduced dynamics should in fact be a good description of the system (http://www.bio-physics.at/wiki/index.php?title=Separation_of_Timescales; also http://www.scholarpedia.org/article/Multiple_scale_analysis). From the figures, we can see that the fast dynamics are the competition between G_wt and G_di, such that the total is roughly only a function of R while they play in between. The difference between the singly and dually-infected situations may simply come down to the initial condition (P) and may be derivable by hand (this is supported by the full model results for singly vs dually-infected cells). If this argument holds up, the reduced model may arguably be the better model and the main text can be simplified to only have one model in the figures and with a simpler description of model fitting. (THAT HAVING BEEN SAID, if the authors can't make this argument work, don't find it as enlightening as I do, or in fact I'm just wrong, the current model is publishable.)

Now Fig 5F is a nice prediction!

**Part III – Minor Issues: Editorial and Data Presentation Modifications**

Reviewer #1: MINOR POINTS:

a. It is not obvious to this reviewer why it is expected that the mutations in DI will not affect the encapsidation efficiency.

b. The last sentence of Results is hard to understand.

c. End of Section 4.1: give all units for the range of values for the initial replication rate (is it 3.02-3.07.10E-2 or 3.02.10E-2-3.07.10E-2?)

Reviewer #2: Line 166: limiting resource are produced at a constant rate lambda, not linear, I suppose.

Page 12: Burst sizes are relative or absolute numbers? If the latter, the values are without units (e.g. line 320).

Page 12, line 334: It may be useful to interpret the results with respect to FLV minimization and not FLV maximization or DVG maximization since reduction of FLV should be the goal with DVG interference/treatments.

Line 682: The formula for g_wt = g_wt^tot – v_di should probably be g_wt = g_wt^tot – v_wt?

Line 693: It should probably be “…Each optimization consisted in minimizing the sum of squared errors between experimental and simulated normalized data…”. Least squares is a method.

Figure 4, line 6: omega is a ratio not a rate.

Supplementary Methods/Description of the models: I’m not sure how the models starting with eq. (13) up to (21) are relevant. The original model is enough as it describes all main determinants of the FLV-DVG dynamics that the authors want to address. And the reduced model is already in the Methods.

Line 1054: instead of “…minimizing the sum of the least square difference between…” should be “…minimizing the sum of squared differences between…”

Reviewer #3: Minor comments

Line 39: I really want an Oxford comma after "apparent"

Line 91: I was hoping for an em-dash saying which two steps... (I had to go back to the abstract to find replication and encapsidation, but it would be nice to be re-reminded and cued up for the results after the intro).

Figure 3 caption: Is "Experimental data and model predictions" the right phrase? The model was fit to this data and it's not an out-of-sample prediction. I would prefer simply "Experimental data and model" although I acknowledge "prediction" is (over)used for this type of figure by others.

Figure S1E: "-1log" is strange notation. Maybe just "-log"

PLOS authors have the option to publish the peer review history of their article (what does this mean?). If published, this will include your full peer review and any attached files.

Reviewer #1: No

Reviewer #2: No

Reviewer #3: No
---

## [Decision Letter · Decision Letter 1]

28 Jul 2021

Dear Raul,

We are pleased to inform you that your manuscript 'Experimental and mathematical insights on the interactions between poliovirus and a defective interfering genome' has been provisionally accepted for publication in PLOS Pathogens.  Good show, Doc!

Best regards,

Bert L. Semler

Guest Editor

PLOS Pathogens

Adolfo García-Sastre

Section Editor

PLOS Pathogens

Kasturi Haldar

Editor-in-Chief

PLOS Pathogens

orcid.org/0000-0001-5065-158X

Michael Malim

Editor-in-Chief

PLOS Pathogens

orcid.org/0000-0002-7699-2064

Reviewer Comments (if any, and for reference):

Reviewer's Responses to Questions

**Part I - Summary**

Reviewer #1: (No Response)

Reviewer #2: I have read the revised version of the manuscript entitled "Experimental and mathematical insights on the interactions between poliovirus and a defective interfering genome" and recommend it for publication as the authors have carefully addressed all my and other reviewers' comments. I have no further suggestions or questions.

**Part II – Major Issues: Key Experiments Required for Acceptance**

Reviewer #1: (No Response)

Reviewer #2: (No Response)

**Part III – Minor Issues: Editorial and Data Presentation Modifications**

Reviewer #1: (No Response)

Reviewer #2: (No Response)

PLOS authors have the option to publish the peer review history of their article (what does this mean?). If published, this will include your full peer review and any attached files.

Reviewer #1: No

Reviewer #2: No

---

## [Editor Report · Acceptance letter]

22 Sep 2021

Dear Dr. Andino,

We are delighted to inform you that your manuscript, "Experimental and mathematical insights on the interactions between poliovirus and a defective interfering genome," has been formally accepted for publication in PLOS Pathogens.

Best regards,

Kasturi Haldar

Editor-in-Chief

PLOS Pathogens

orcid.org/0000-0001-5065-158X

Michael Malim

Editor-in-Chief

PLOS Pathogens

orcid.org/0000-0002-7699-2064